# TRAINING-FREE LINEAR IMAGE INVERSION VIA FLOWS

## ABSTRACT

Solving inverse problems without any training involves using a pretrained generative model and making appropriate modifications to the generation process to avoid finetuning of the generative model. While recent methods have explored the use of diffusion models, they still require the manual tuning of many hyperparameters for different inverse problems. In this work, we propose a training-free method for solving linear inverse problems by using pretrained flow models, leveraging the simplicity and efficiency of Flow Matching models, using theoretically-justified weighting schemes, and thereby significantly reducing the amount of manual tuning. In particular, we draw inspiration from two main sources: adopting prior gradient correction methods to the flow regime, and a solver scheme based on conditional Optimal Transport paths. As pretrained diffusion models are widely accessible, we also show how to practically adapt diffusion models for our method. Empirically, our approach requires no problem-specific tuning across an extensive suite of noisy linear inverse problems on high-dimensional datasets, ImageNet-64/128 and AFHQ-256, and we observe that our flow-based method for solving inverse problems significantly improves upon closely-related diffusion-based methods.

## 1 INTRODUCTION

Solving an inverse problem involves recovering a clean signal from noisy measurements generated by a known degradation model. Many interesting image processing tasks can be cast as an inverse problem. Some instances of these problems are super-resolution, inpainting, deblurring, colorization, denoising etc. Diffusion models or score-based generative models (Sohl-Dickstein et al., 2015; Ho

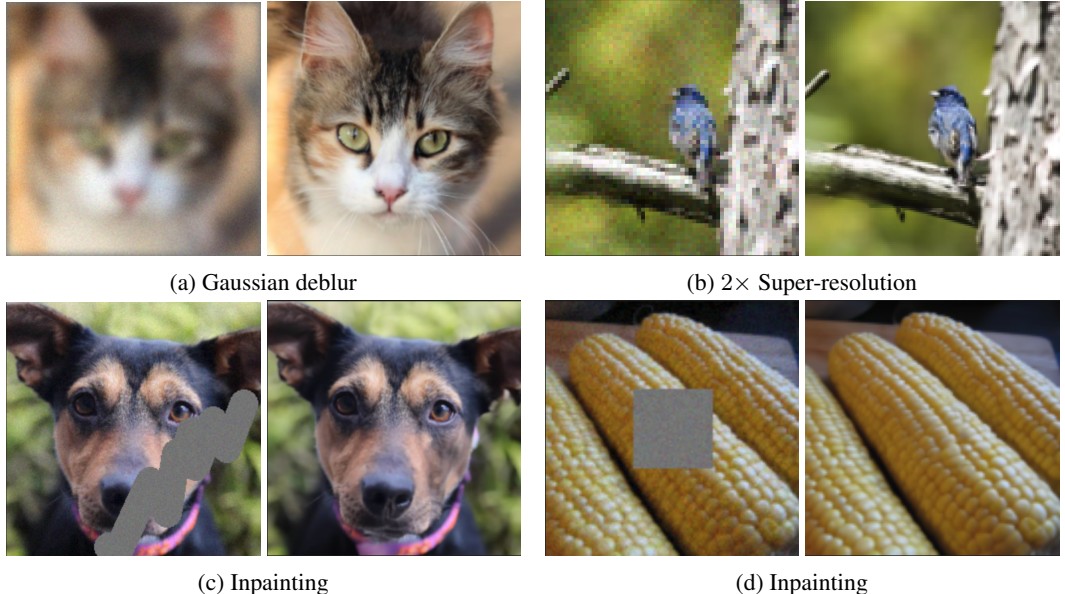

(a) Gaussian deblur        (b) 2× Super-resolution

(c) Inpainting        (d) Inpainting

Figure 1: Corrected images by solving linear inverse problems with flow models. For each pair of images, we show the noisy measurement (left) and the reconstruction (right).

et al., 2020; Song & Ermon, 2019; Song et al., 2021c) have emerged as a leading family of generative models for solving inverse problems for images (Saharia et al., 2022b;a; Wang et al., 2022; Chung et al., 2022a; Song et al., 2022; Mardani et al., 2023). However, sampling with diffusion models is known to be slow, and the quality of generated images is affected by the curvature of SDE/ODE solution trajectories (Karras et al., 2022). While Karras et al. (2022) observed ODE sampling for image generation could produce better results, sampling via SDE is still common for solving inverse problems, whereas ODE sampling has been rarely considered, perhaps due to the use of diffusion probability paths.

Continuous Normalizing Flow (CNF) (Chen et al., 2018b) trained with Flow Matching (Lipman et al., 2022) has been recently proposed as a powerful alternative to diffusion models. CNF (hereafter denoted flow model) has the ability to model arbitrary probability paths, and includes diffusion probability paths as a special case. Of particular interest to us are Gaussian probability paths that correspond to optimal transport (OT) displacement (McCann, 1997). Recent works (Lipman et al., 2022; Albergo & Vanden-Eijnden, 2022; Liu et al., 2022; Shaul et al., 2023) have shown that these conditional OT probability paths are straighter than diffusion paths, which results in faster training and sampling with these models. Due to these properties, conditional OT flow models are an appealing alternative to diffusion models for solving inverse problems.

In this work, we introduce a training-free method to utilize pretrained flow models for solving linear inverse problems. Our approach adds a correction term to the vector field that takes into account knowledge from the degradation model. Specifically, we introduce an algorithm that incorporates $\Pi$GDM (Song et al., 2022) gradient correction to flow models. Given the wide availability of pretrained diffusion models, we also present a way to convert these models to arbitrary paths for our sampling procedure. Empirically, we observe images restored via a conditional OT path consistently exhibit perceptual quality better than that achieved by the model's original diffusion path as well as recently proposed diffusion approaches, such as $\Pi$GDM (Song et al., 2022) and RED-Diff (Mardani et al., 2023), across all linear inverse problems. To summarize, our key contributions are:

- We present a training-free approach for solving linear inverse problems with pretrained conditional OT flow models that adapts the $\Pi$GDM correction, proposed for diffusion models, to flow sampling.
- We offer a way to convert between flow models and diffusion models, enabling the use of pretrained continuous-time diffusion models for conditional OT sampling, and vice versa.
- We demonstrate that images restored via our algorithm using conditional OT probability paths have perceptual quality that is on par with, or better than that achieved by diffusion probability paths, and other recent methods like $\Pi$GDM and RED-Diff.

## 2 PRELIMINARIES

We introduce relevant background knowledge and notation from conditional diffusion and flow modeling, as well as training-free inference with diffusion models.

**Notation.** Both diffusion and flow models consider two distinct processes indexed by time between $[0, 1]$ that convert data to noise and noise to data. Here, we follow the temporal notation used in prior work (Lipman et al., 2022) where the distribution at $t = 1$ is the data distribution and $t = 0$ is a standard Gaussian distribution. Note that this is opposite of the prevalent notation used in diffusion model literature (Song & Ermon, 2019; Ho et al., 2020; Song et al., 2021a;c). We let $\boldsymbol{x}_t$ denote the value at time $t$, without regard to which process (*i.e.,* diffusion or flow) it was drawn from. The probability density for the data to noise process is denoted $q$ and the parameterized probability density for the noise to data process is denoted $p_\theta$. Expectations with respect to $q$ are denoted via $\mathbb{E}_q$. We generally keep function arguments of $t$ implicit (i.e. $f(\boldsymbol{x}_t, t)$ is informally written as $f(\boldsymbol{x}_t)$.)

**Conditional diffusion models.** Conditional diffusion uses a class of processes $q$, and defines data to noise process $p_\theta$ as a Markov process that in continuous time obeys a stochastic differential equation (SDE) (Song & Ermon, 2019; Ho et al., 2020; Song et al., 2021c). The parameters of $p_\theta$ are learned via minimizing a regression loss with respect to $\widehat{\boldsymbol{x}_1}$

$$L_{\text{diffusion}}(\widehat{\boldsymbol{x}_1}, q) = \int_0^1 w(t)\mathbb{E}_q[(\widehat{\boldsymbol{x}_1}(\boldsymbol{x}_t, \boldsymbol{y}) - \boldsymbol{x}_1)^2]dt \tag{1}$$

where $w(t)$ are positive weights (Kingma et al., 2021; Song et al., 2021b; Kingma & Gao, 2023), $\boldsymbol{y}$ is conditioning (i.e. a noisy image), and $\boldsymbol{x}_1$ is noiseless data. The optimal solution for $\widehat{\boldsymbol{x}_1}$ is $\mathbb{E}_q[\boldsymbol{x}_1|\boldsymbol{x}_t, \boldsymbol{y}]$, and hence we refer to $\widehat{\boldsymbol{x}_1}$ as a denoiser. Many equivalent parameterizations exist and have known conversions to denoising. Sampling using $p_\theta$ proceeds via starting from $\boldsymbol{x}_0 \sim p_\theta(\boldsymbol{x}_0|\boldsymbol{y})$ and integrating the SDE to $t = 1$. If $p_\theta(\boldsymbol{x}_0|\boldsymbol{y}) = q(\boldsymbol{x}_0|\boldsymbol{y})$, the SDE is integrated exactly, and $\widehat{\boldsymbol{x}_1} = \mathbb{E}_q[\boldsymbol{x}_1|\boldsymbol{x}_t, \boldsymbol{y}]$, the resulting $\boldsymbol{x}_1 \sim q(\boldsymbol{x}_1|\boldsymbol{y})$.

**Conditional flow models**   Alternatively, continuous normalizing flow models (Chen et al., 2018b) define the data generation process through an ODE. This leads to simpler formulations and does not introduce extra noise during intermediate steps of sample generation. Recently, simulation-free training algorithms have been designed specifically for such models (Lipman et al., 2022; Liu et al., 2022; Albergo & Vanden-Eijnden, 2022), an example being the Conditional Flow Matching loss (Lipman et al., 2022),

$$L_{\text{cfm}}(\widehat{\boldsymbol{v}}, q) = \int_0^1 \mathbb{E}_q \left[ \left( \widehat{\boldsymbol{v}}(\boldsymbol{x}_t, \boldsymbol{y}) - \mathbb{E}_q \left[ \frac{d\boldsymbol{x}_t}{dt} \middle| \boldsymbol{x}_t, \boldsymbol{y}, \boldsymbol{x}_1 \right] \right)^2 \right] dt. \tag{2}$$

where $\widehat{\boldsymbol{v}}(\boldsymbol{x}_t, \boldsymbol{y})$ denotes a parameterized vector field defining the ODE

$$\frac{d\boldsymbol{x}_t}{dt} = \widehat{\boldsymbol{v}}(\boldsymbol{x}_t, \boldsymbol{y}). \tag{3}$$

If trained perfectly, the marginal distributions of $\boldsymbol{x}_t$, denoted $p_\theta(\boldsymbol{x}_t|\boldsymbol{y})$, will match the marginal distributions of $q(\boldsymbol{x}_t|\boldsymbol{y})$. Hence sampling from $q(\boldsymbol{x}_t|\boldsymbol{y})$ involves starting from an initial value $\boldsymbol{x}_{t'} \sim q(\boldsymbol{x}_{t'}|\boldsymbol{y})$ and integrating the ODE from $t'$ to $t$. Typically, one samples from $t' = 0$ since $q(\boldsymbol{x}_0|\boldsymbol{y})$ is a tractable distribution. Furthermore, for general Gaussian probability paths $q(\boldsymbol{x}_t|\boldsymbol{x}_1, \boldsymbol{y}) = \mathcal{N}(\mu_t(\boldsymbol{y}, \boldsymbol{x}_1), \sigma_t(\boldsymbol{y}, \boldsymbol{x}_1)^2 \boldsymbol{I})$, one can set (Lipman et al., 2022)

$$\mathbb{E}_q \left[ \frac{d\boldsymbol{x}_t}{dt} \middle| \boldsymbol{x}_t, \boldsymbol{y}, \boldsymbol{x}_1 \right] = \frac{d\mu_t}{dt} + \frac{d\sigma_t}{dt} \left( \frac{\boldsymbol{x}_t - \mu_t}{\sigma_t} \right). \tag{4}$$

**Gaussian probability paths.**   The time-dependent distributions $q(\boldsymbol{x}_t|\boldsymbol{y}, \boldsymbol{x}_1)$ are referred to as conditional probability paths. We focus on the class of affine Gaussian probability paths of the form

$$q(\boldsymbol{x}_t|\boldsymbol{y}, \boldsymbol{x}_1) = q(\boldsymbol{x}_t|\boldsymbol{x}_1) = \mathcal{N}(\alpha_t \boldsymbol{x}_1, \sigma_t^2 \boldsymbol{I}) \tag{5}$$

where non-negative $\alpha_t$ and $\sigma_t$ are monotonically increasing and decreasing respectively. This class includes the probability paths for conditional diffusion as well as the conditional Optimal Transport (OT) path (Lipman et al., 2022), where $\alpha_t = t$ and $\sigma_t = 1 - t$. The conditional OT path used by flow models has been demonstrated to have good empirical properties, including faster inference and better sampling in practice, and has theoretical support in high-dimensions (Shaul et al., 2023). As emphasized in Lin et al. (2023), a desirable property for probability paths, obeyed by conditional OT but not commonly used diffusion paths, is to ensure $q(\boldsymbol{x}_0|\boldsymbol{y})$ is known (i.e. $\mathcal{N}(0, \boldsymbol{I})$), as otherwise one cannot exactly sample $\boldsymbol{x}_0$ which can add substantial error.

**Training-free conditional inference using unconditional diffusion.**   Given pretrained *unconditional* diffusion models that are trained to approximate $\mathbb{E}_q[\boldsymbol{x}_1|\boldsymbol{x}_t]$, training-free approaches aim to approximate $\mathbb{E}_q[\boldsymbol{x}_1|\boldsymbol{x}_t, \boldsymbol{y}]$. Under Gaussian probability paths, the two terms are related as Tweedie's identity (Robbins, 1992) expresses $\mathbb{E}_q[\boldsymbol{x}_1|\boldsymbol{x}_t, \boldsymbol{y}] = (\boldsymbol{x}_t + \sigma_t^2 \nabla_{\boldsymbol{x}_t} \ln q(\boldsymbol{x}_t|\boldsymbol{y}))/\alpha_t$. Applying this identity (twice for both $\mathbb{E}_q[\boldsymbol{x}_1|\boldsymbol{x}_t, \boldsymbol{y}]$ and $\mathbb{E}_q[\boldsymbol{x}_1|\boldsymbol{x}_t]$) and simplifying gives

$$\mathbb{E}_q[\boldsymbol{x}_1|\boldsymbol{x}_t, \boldsymbol{y}] = \mathbb{E}_q[\boldsymbol{x}_1|\boldsymbol{x}_t] + \frac{\sigma_t^2}{\alpha_t} \nabla_{\boldsymbol{x}_t} \ln q(\boldsymbol{y}|\boldsymbol{x}_t). \tag{6}$$

Following Eq. 6, past approaches (*e.g.,* Chung et al. (2022a); Song et al. (2022)) have used the pretrained model for the first term and approximated the second intractable term to produce an approximate $\widehat{\boldsymbol{x}_1}(\boldsymbol{x}_t, \boldsymbol{y})$. Diffusion posterior sampling (DPS) (Chung et al., 2022a) proposed to approximate $q(\boldsymbol{y}|\boldsymbol{x}_t)$ via $q(\boldsymbol{y}|\boldsymbol{x}_1 = \widehat{\boldsymbol{x}_1}(\boldsymbol{x}_t))$. Later, Pseudo-inverse Guided Diffusion Models (ΠGDM) (Song et al., 2022) improved upon DPS for linear noisy observations where $\boldsymbol{y} = \boldsymbol{A}\boldsymbol{x} + \sigma_y \epsilon$, where $\boldsymbol{A}$ is some measurement matrix and $\epsilon \sim \mathcal{N}(0, \boldsymbol{I})$, by approximating $q(\boldsymbol{y}|\boldsymbol{x}_t)$ as $\mathcal{N}(\boldsymbol{A}\widehat{\boldsymbol{x}_1}(\boldsymbol{x}_t), \sigma_{\boldsymbol{y}}^2 \boldsymbol{I} + r_t^2 \boldsymbol{A}\boldsymbol{A}^T)$, derived via first approximating $q(\boldsymbol{x}_1|\boldsymbol{x}_t)$ as $\mathcal{N}(\widehat{\boldsymbol{x}_1}(\boldsymbol{x}_t), r_t^2 \boldsymbol{I})$. ΠGDM also suggested adaptive weighting, replacing $\sigma_t^2/\alpha_t$ with another function of time to account for the approximation.

## 3 Solving Linear Inverse Problems without Training via Flows

In the standard setup of a linear inverse problem, we observe measurements $\boldsymbol{y} \in \mathbb{R}^n$ such that

$$\boldsymbol{y} = \boldsymbol{A}\boldsymbol{x}_1 + \epsilon \tag{7}$$

where $\boldsymbol{x}_1 \in \mathbb{R}^m$ is drawn from an unknown data distribution $q(\boldsymbol{x}_1)$, $\boldsymbol{A} \in \mathbb{R}^{n \times m}$ is a known measurement matrix, and $\epsilon \sim \mathcal{N}(0, \sigma_y^2 \boldsymbol{I})$ is unknown *i.i.d.* Gaussian noise with known standard deviation $\sigma_y$. Given a pretrained flow model with $\widehat{\boldsymbol{v}}(\boldsymbol{x}_t)$ that can sample from $q(\boldsymbol{x}_1)$, and measurements $\boldsymbol{y}$, our goal is to produce clean samples from the posterior $q(\boldsymbol{x}_1|\boldsymbol{y}) \propto q(\boldsymbol{y}|\boldsymbol{x}_1)q(\boldsymbol{x}_1)$ without training a problem-specific conditional flow model defined by $\widehat{\boldsymbol{v}}(\boldsymbol{x}_t, \boldsymbol{y})$. Proofs are in Appendix A.

### 3.1 Converting between diffusion models and flow models

While our proposed algorithm is based on flow models, this poses no limitation on needing pretrained flow models. We first take a brief detour outside of inverse problems to demonstrate continuous-time diffusion models parameterized by $\widehat{\boldsymbol{x}_1}$ can be converted to flow models $\widehat{\boldsymbol{v}}$ that have Gaussian probability paths described by Eq. 5. We separate training and sampling such that one can train with Gaussian probability path $q'$ and then perform sampling with a different Gaussian probability path $q$. Equivalent expressions to this subsection have been derived in Karras et al. (2022), leveraging a more general conversion from SDE to probability flow ODE from Song et al. (2021c). Karras et al. (2022) similarly separate training and sampling for Gaussian probability paths, and demonstrate that using alternative sampling paths was beneficial. Our derivations avoid the SDE via a flow perspective and exposes a subtlety when swapping probability paths using pretrained models.

**Lemma 1.** *For Gaussian probability path $q$ given by Eq. 5, the optimal solution for $\widehat{\boldsymbol{v}}(\boldsymbol{x}_t, \boldsymbol{y})$ is known given $\mathbb{E}_q[\boldsymbol{x}_1|\boldsymbol{x}_t, \boldsymbol{y}]$, and vice versa.*

In particular, a diffusion model's denoiser $\widehat{\boldsymbol{x}_1}(\boldsymbol{x}_t, \boldsymbol{y})$ trained using Gaussian probability path $q$ can be interchanged with a flow model's $\widehat{\boldsymbol{v}}(\boldsymbol{x}_t, \boldsymbol{y})$ with the same $q$ via

$$\widehat{\boldsymbol{v}} = \left(\alpha_t \frac{d \ln(\alpha_t/\sigma_t)}{dt}\right)\widehat{\boldsymbol{x}_1} + \frac{d \ln \sigma_t}{dt}\boldsymbol{x}_t. \tag{8}$$

Furthermore, $\widehat{\boldsymbol{x}_1}(\boldsymbol{x}_t, \boldsymbol{y})$ trained under $q'$ can be used for another Gaussian $q$ during sampling.

**Lemma 2.** *Consider two Gaussian probability paths $q$ and $q'$ defined by Eq. 5 with $\alpha_t$, $\sigma_t$ and $\alpha'_t$, $\sigma'_t$ respectively. Define $t'(t)$ as the unique solution to $\frac{\sigma_t}{\alpha_t} = \frac{\sigma'_{t'}}{\alpha'_{t'}}$ when it exists for given t. Then*

$$\mathbb{E}_q[\boldsymbol{x}_1|\boldsymbol{x}_t, \boldsymbol{y}] = \mathbb{E}_{q'}[\boldsymbol{x}_1|\boldsymbol{X}'_{t'(t)} = \alpha'_{t'(t)}\boldsymbol{x}_t/\alpha_t, \boldsymbol{y}]. \tag{9}$$

So if $\widehat{\boldsymbol{x}_1}$ was trained under $q'$ it can be used for sampling under $q$ via evaluating at $\widehat{\boldsymbol{x}_1}(\alpha'_{t'(t)}\boldsymbol{x}_t/\alpha_t, t'(t), \boldsymbol{y})$ (with explicit time for clarity) whenever $t'(t)$ exists. In particular, if pretrained denoiser was trained with $q'$ and we perform conditional OT sampling, we utilize

$$t'(t) = \text{SNR}_{q'}^{-1}(\text{SNR}_q(t)) = \text{SNR}_{q'}^{-1}\left(\frac{t}{1-t}\right). \tag{10}$$

where signal-to-noise ratio $\text{SNR}(t) = \alpha_t/\sigma_t$. The main avenue for non-existence for $t'(t)$ is if the model under $q'$ is trained using a minimum SNR above zero, which induces a minimum $t$ for which $t'(t)$ exists. When a minimum $t$ exists, we can only perform sampling with $q$ starting from $\boldsymbol{x}_t \sim q(\boldsymbol{x}_t|\boldsymbol{y})$. Approximating this sample is entirely analogous to approximating $\boldsymbol{x}_0 \sim q'(\boldsymbol{x}_0|\boldsymbol{y})$. This error already exists for $q'$ because $q'(\boldsymbol{x}_0|\boldsymbol{y})$ is not $\mathcal{N}(0, \boldsymbol{I})$ unless $q'$ is trained to zero SNR. An initialization problem cannot be avoided if $q'$ has limited SNR range by switching paths to $q$.

### 3.2 Correcting the vector field of flow models

To solve linear inverse problems without any training via flow models, we derive an expression similar to Eq. 6 that relates conditional vector fields under Gaussian probability paths to unconditional vector fields.

**Theorem 1.** *Let $q$ be a Gaussian probability path described by Eq. 5. Assume we observe $\boldsymbol{y} \sim q(\boldsymbol{y}|\boldsymbol{x}_1)$ for arbitrary $q(\boldsymbol{y}|\boldsymbol{x}_1)$ and $v(\boldsymbol{x}_t)$ is a vector field enabling sampling $\boldsymbol{x}_t \sim q(\boldsymbol{x}_t)$. Then a vector field $v(\boldsymbol{x}_t, \boldsymbol{y})$ enabling sampling $\boldsymbol{x}_t \sim q(\boldsymbol{x}_t|\boldsymbol{y})$ can be written*

$$v(\boldsymbol{x}_t, \boldsymbol{y}) = v(\boldsymbol{x}_t) + \sigma_t^2 \frac{d \ln(\alpha_t/\sigma_t)}{dt} \nabla_{\boldsymbol{x}_t} \ln q(\boldsymbol{y}|\boldsymbol{x}_t). \tag{11}$$

We turn Theorem 1 into an training-free algorithm for solving linear inverse using flows by adapting ΠGDM's approximation. In particular, given $\widehat{\boldsymbol{v}}(\boldsymbol{x}_t)$ (or $\widehat{\boldsymbol{x}_1}(\boldsymbol{x}_t)$), our approximation will be

$$\widehat{\boldsymbol{v}}(\boldsymbol{x}_t, \boldsymbol{y}) = \widehat{\boldsymbol{v}}(\boldsymbol{x}_t) + \sigma_t^2 \frac{d \ln(\alpha_t/\sigma_t)}{dt} \gamma_t \nabla_{\boldsymbol{x}_t} \ln q^{app}(\boldsymbol{y}|\boldsymbol{x}_t), \tag{12}$$

where following ΠGDM terminology, we refer to $\gamma_t = 1$ as unadaptive and other choices as adaptive weights. In general, we view adaptive weights $\gamma_t \neq 1$ as an adjustment for error in $q^{app}(\boldsymbol{y}|\boldsymbol{x}_t)$.

For $q^{app}(\boldsymbol{y}|\boldsymbol{x}_t)$, we generalize ΠGDM to any Gaussian probability path described by Eq. 5 via updating $r_t^2$. We still have $q^{app}(\boldsymbol{y}|\boldsymbol{x}_t)$ is $\mathcal{N}(\boldsymbol{A}\widehat{\boldsymbol{x}_1}(\boldsymbol{x}_t), \sigma_y^2 I + r_t^2 \boldsymbol{A}\boldsymbol{A}^\top)$ when $q(\boldsymbol{x}_1|\boldsymbol{x}_t)$ is approximated as $\mathcal{N}(\widehat{\boldsymbol{x}_1}(\boldsymbol{x}_t), r_t^2 I)$. We choose $r_t^2$ by following ΠGDM's derivation, noting that if $q(\boldsymbol{x}_1)$ is $\mathcal{N}(0, \boldsymbol{I})$ and $q(\boldsymbol{x}_t|\boldsymbol{x}_1)$ is $\mathcal{N}(\alpha_t \boldsymbol{x}_1, \sigma_t^2 I)$, then

$$r_t^2 = \frac{\sigma_t^2}{\sigma_t^2 + \alpha_t^2}. \tag{13}$$

When $\alpha_t = 1$, we recover ΠGDM's $r_t^2$ as expected under their Variance-Exploding path specification.

**Starting flow sampling at time $t > 0$.** Initializing conditional diffusion model sampling at $t > 0$ has been proposed by Chung et al. (2022c). For flows, we similarly want $\boldsymbol{x}_t \sim q(\boldsymbol{x}_t|\boldsymbol{y})$ at initialization time $t$. In our experiments, we examine (approximately) initializing at different times $t > 0$ using

$$\boldsymbol{x}_t = \alpha_t \boldsymbol{y} + \sigma_t \epsilon \tag{14}$$

for $\epsilon \sim \mathcal{N}(0, \boldsymbol{I})$ when $\boldsymbol{y}$ is the correct shape. For super-resolution, we use nearest-neighbor interpolation on $\boldsymbol{y}$ instead. We also consider using $\boldsymbol{A}^\dagger y$ as an ablation in the Appendix C (where $\boldsymbol{A}^\dagger$ is the pseudo-inverse of $\boldsymbol{A}$ (Song et al., 2022)). We may be forced to use this initialization for flow sampling due to converting a diffusion model not trained to zero SNR. However as shown in (Chung et al., 2022c) for diffusion, this initialization can improve results more generally. Conceptually, if the resulting $\boldsymbol{x}_t$ is closer to $\boldsymbol{x}_t \sim q(\boldsymbol{x}_t|\boldsymbol{y})$ than achieved via starting from an earlier time $t'$ and integrating, then this initialization can result in less overall error.

**Algorithm summary.** Putting this altogether, our proposed approach using flow sampling and conditional OT probability paths is succinctly summarized in Algorithm 1, derived via inserting $\alpha_t = t$ and $\sigma_t = 1 - t$. Unlike ΠGDM, we propose unadaptive weights $\gamma_t = 1$. By default, we set initialization time $t = 0.2$. The algorithm therefore has no additional hyperparameters to tune over traditional diffusion or flow sampling. In Appendix B, we detail our algorithm for other Gaussian probability paths, and the equivalent formulation when a pretrained vector field is available instead.

---

**Algorithm 1** Solving linear inverse problems via flows using conditional OT probability path

---

**Require:** Pretrained denoiser $\widehat{\boldsymbol{x}_1}(\boldsymbol{x}_t)$ converted to conditional OT probability path using Section 3.1, noisy measurement $\boldsymbol{y}$, measurement matrix $\boldsymbol{A}$, initial time $t$, and std $\sigma_y$

1: Initialize $\boldsymbol{x}_t = t\boldsymbol{y} + (1-t)\epsilon$, where $\epsilon \sim \mathcal{N}(0, \boldsymbol{I})$          ▷ Initialize $\boldsymbol{x}_t$, Eq. (14)

2:   $\boldsymbol{z}_t = \boldsymbol{x}_t$

3: **for** each time step $t'$ of ODE integration **do**          ▷ Integrate ODE from $t' = t$ to 1.

4:     $r_{t'}^2 = \frac{(1-t')^2}{t'^2 + (1-t')^2}$          ▷ Value of $r_t^2$ from Eq. (13)

5:     $\widehat{\boldsymbol{v}} = \frac{\widehat{\boldsymbol{x}_1}(\boldsymbol{z}_{t'}) - \boldsymbol{z}_{t'}}{1 - t'}$          ▷ Convert $\widehat{\boldsymbol{x}_1}$ to vector field, Eq. (8)

6:     $\boldsymbol{g} = (\boldsymbol{y} - \boldsymbol{A}\widehat{\boldsymbol{x}_1})^\top (r_{t'}^2 \boldsymbol{A}\boldsymbol{A}^\top + \sigma_y^2 \boldsymbol{I})^{-1} \boldsymbol{A} \frac{\partial \widehat{\boldsymbol{x}_1}}{\partial \boldsymbol{z}_{t'}}$          ▷ ΠGDM correction

7:     $\widehat{\boldsymbol{v}}_{\text{corrected}} = \widehat{\boldsymbol{v}} + \frac{1-t'}{t'} \boldsymbol{g}$          ▷ Correct unconditional vector field $\widehat{\boldsymbol{v}}$, Eq. (12)

8: **end for**

---

## 4 EXPERIMENTS

**Datasets.** We verify the effectiveness of our proposed approach on three datasets: face-blurred ImageNet $64 \times 64$ and $128 \times 128$ (Deng et al., 2009; Russakovsky et al., 2015; Yang et al., 2022), and AnimalFacesHQ (AFHQ) $256 \times 256$ (Choi et al., 2020). We report our results on 10K randomly sampled images from validation split of ImageNet, and 1500 images from test split of AFHQ.

**Tasks.** We report results on the following linear inverse problems: inpainting (center-crop), Gaussian deblurring, super-resolution, and denoising. The exact details of the measurement operators are: 1) For inpainting, we use centered mask of size $20 \times 20$ for ImageNet-64, $40 \times 40$ for ImageNet-128, and $80 \times 80$ for AFHQ. In addition, for images of size $256 \times 256$, we also use free-form masks simulating brush strokes similar to the ones used in Saharia et al. (2022a); Song et al. (2022). 2) For super-resolution, we apply bicubic interpolation to downsample images by $4\times$ for datasets that have images with resolution $256 \times 256$ and downsample images by $2\times$ otherwise. 3) For Gaussian deblurring, we apply Gaussian blur kernel of size $61 \times 61$ with intensity value 1 for ImageNet-64 and ImageNet-128, and $61 \times 61$ with intensity value 3 for AFHQ. 4) For denoising, we add *i.i.d.* Gaussian noise with $\sigma_y = 0.05$ to the images. For tasks besides denoising, we consider *i.i.d.* Gaussian noise with $\sigma_y = 0$ and 0.05 to the images. Images $\boldsymbol{x}_1$ are normalized to range $[-1, 1]$.

**Implementation details.** We trained our own continuous-time conditional VP-SDE model, and conditional Optimal Transport (conditional OT) flow model from scratch on the above datasets following the hyperparameters and training procedure outlined in Song et al. (2021c) and Lipman et al. (2022). These models are conditioned on class labels, not noisy images. All derivations hold with class label $c$ since $q(\boldsymbol{y}|c, \boldsymbol{x}_1) = q(\boldsymbol{y}|\boldsymbol{x}_1)$ (i.e. the noisy image is independent of class label given the image). We use the open-source implementation of the Euler method provided in torchdiffeq library (Chen, 2018) to solve the ODE in our experiments. Our choice of Euler is intentionally simple, as we focus on flow sampling with the conditional OT path, and not on the choice of ODE solver.

**Metrics.** We follow prior works (Chung et al., 2022a; Kawar et al., 2022) and report Fréchet Inception Distance (FID) (Heusel et al., 2017), Learned Perceptual Image Patch Similarity (LPIPS) (Zhang et al., 2018), peak signal-to-noise ratio (PSNR), and structural similarity index (SSIM). We use open-source implementations of these metrics in the TorchMetrics library (Detlefsen et al., 2022).

**Methods and baselines.** We use our two pretrained model checkpoints— a conditional OT flow model and continuous VP-SDE diffusion model, and perform flow sampling with both conditional OT and Variance-Preserving (VP) paths, labeling our methods as OT-ODE and VP-ODE respectively. Because qualitative results are identical and quantitative results similar, we only include the VP-SDE diffusion model in the main text, and include the conditional OT flow model in Appendix D. We compare our OT-ODE and VP-ODE methods against ΠGDM (Song et al., 2022) and RED-Diff (Mardani et al., 2023) as relevant baselines. We selected these baselines because they achieve state-of-the-art performance in solving linear inverse problems using diffusion models. The code for both baseline methods is available on github, and we make minimal changes while reimplementing these methods in our codebase. A fair comparison between methods requires considering the number of function evaluations (NFEs) used during sampling. We utilize at most 100 NFEs for our OT-ODE and VP-ODE sampling (see Appendix D), and utilize 100 for ΠGDM as recommended in Song et al. (2022). We allow RED-Diff 1000 NFEs since it does not require gradients of $\widehat{\boldsymbol{x}_1}$. For OT-ODE following Algorithm 1, we use $\gamma_t = 1$ and initial $t = 0.2$ for all datasets and tasks. For VP-ODE following Algorithm 2 in the Appendix, we use $\gamma_t = \sqrt{\frac{\alpha_t}{\alpha_t^2 + \sigma_t^2}}$ and initial $t = 0.4$ for all datasets and tasks. Ablations of these mildly tuned hyperparameters are shown in Appendix C. We extensively tuned hyperparameters for RED-Diff and ΠGDM as described in Appendix E, including different hyperparameters per dataset and task.

### 4.1 EXPERIMENTAL RESULTS

We report quantitative results for the VP-SDE model, across all datasets and linear measurements, in Figure 2 for $\sigma_y = 0.05$, and in Figure 12 within Appendix D.1 for $\sigma_y = 0$. Additionally, we report results for the conditional OT flow model in Figure 14 and Figure 13 for $\sigma_y = 0.05$ and $\sigma_y = 0$,

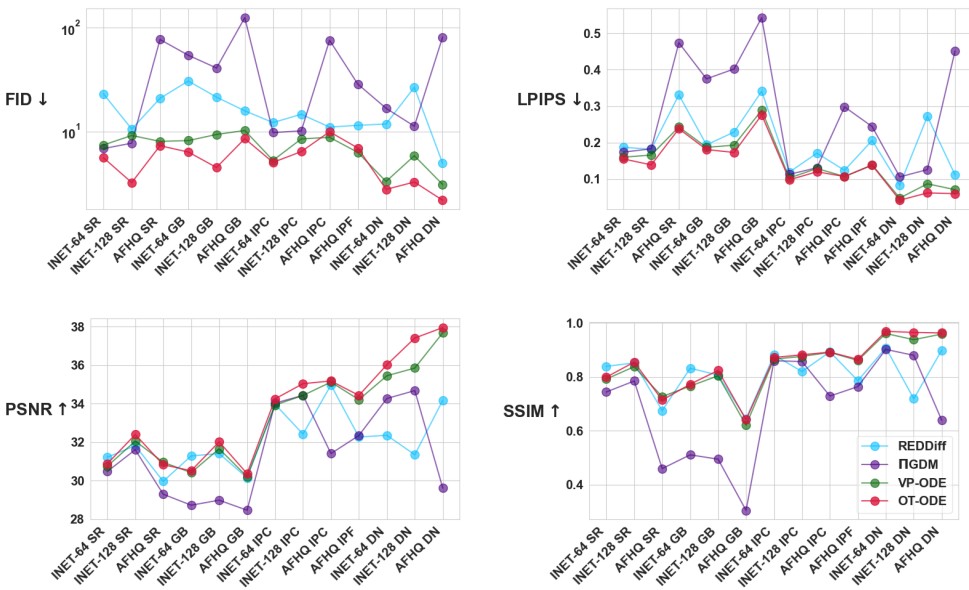

Figure 2: Quantitative evaluation of pretrained VP-SDE model for solving linear inverse problems on super-resolution (SR), gaussian deblurring (GB), inpainting with centered (IPC) and freeform mask (IPF), and denoising (DN) with $\sigma_y = 0.05$. We present results on face-blurred ImageNet-64 (INET-64), face-blurred ImageNet-128 (INET-128), and AFHQ.

respectively, in Appendix D.1. Exact numerical values for all the metrics across all datasets and tasks can be found in Appendix D.

**Gaussian Deblurring.** We report qualitative results for the VP-SDE model in Figure 3 and for the conditional OT flow (cond-OT) model in Figure 15. We observe that for both the VP-SDE model and the cond-OT model, OT-ODE and VP-ODE outperform ΠGDM and RED-Diff, both qualitatively and quantitatively, across all datasets for both $\sigma_y = 0.05$ and $\sigma_y = 0$. As shown in Figure 3 and 15, ΠGDM tends to sharpen the images, which sometimes results in unnatural textures in the images. Further, we also observe some unnatural textures and background noise with RED-Diff for $\sigma_y = 0.05$.

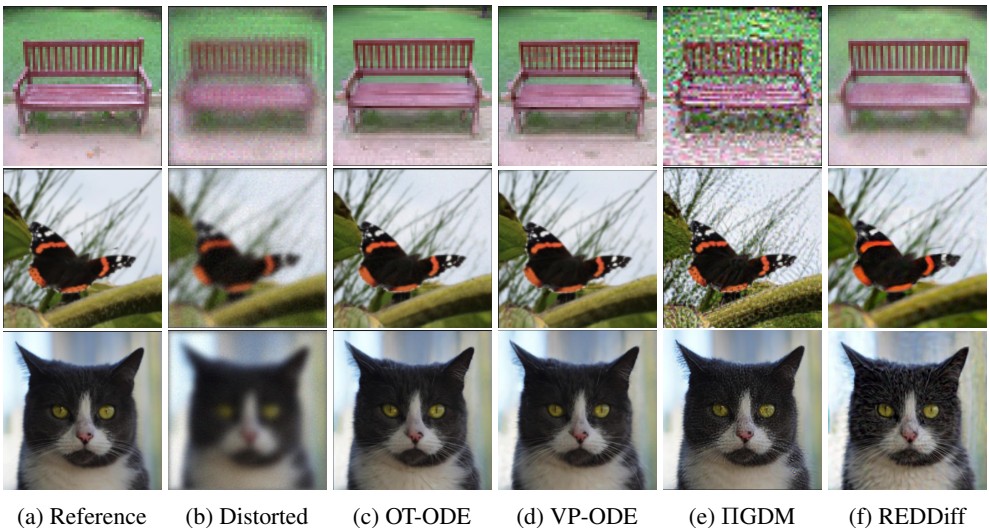

|(a) Reference|(b) Distorted|(c) OT-ODE|(d) VP-ODE|(e) ΠGDM|(f) REDDiff|

Figure 3: Results for Gaussian deblurring with VP-SDE model and $\sigma_y = 0.05$ for (**first row**) ImageNet-64, (**second row**) ImageNet-128, and (**third row**) AFHQ.

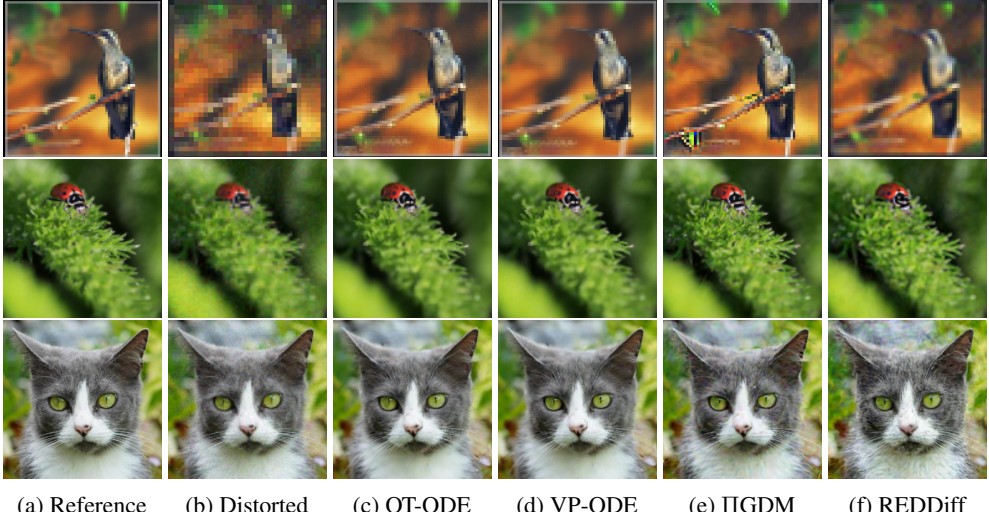

|  (a) Reference | (b) Distorted | (c) OT-ODE | (d) VP-ODE | (e) ΠGDM | (f) REDDiff |

Figure 4: Results for super-resolution with VP-SDE model and $\sigma_y = 0.05$ for (**first row**) ImageNet-64 $2\times$, (**second row**) ImageNet-128 $2\times$, and (**third row**) AFHQ $4\times$.

**Super-resolution.** We report qualitative results for the VP-SDE model in Figure 4 and for the cond-OT model in Figure 16. OT-ODE sampling consistently achieves better FID, LPIPS and PSNR metrics compared to other methods with both VP-SDE model and cond-OT model for $\sigma_y = 0.05$ (See Figure 2 and 14). Similar to Gaussian deblurring, ΠGDM tends to produce sharper edges. This is certainly desirable to achieve good super-resolution, but sometimes this results in unnatural textures in the images (See Figure 4). RED-Diff for $\sigma_y = 0.05$ gives slightly blurry images. In our experiments, we observe RED-Diff is sensitive to the values of $\sigma_y$, and we get good quality results for smaller values of $\sigma_y$, but the performance deteriorates with increase in value of $\sigma_y$. For noiseless case, as shown in Figure 22 and Figure 25, all the methods achieve comparable performance.

**Inpainting.** For centered mask inpainting, OT-ODE sampling outperforms ΠGDM and RED-Diff in terms of LPIPS, PSNR and SSIM across all datasets at $\sigma_y = 0.05$ for both the VP-SDE and cond-OT model. Regarding FID, OT-ODE performs comparably to or better than VP-SDE (See Figure 2 and 14). Similar observations hold true for inpainting with freeform mask on AFHQ. We present qualitative results for the VP-SDE model in Figure 5 and the cond-OT model in Figure 17. As evident in these images, OT-ODE sampling results in more semantically meaningful inpainting (for instance, the shape of bird's neck, and shape of hot-dog bread in Figure 5). In contrast, the inpainted regions generated by RED-Diff tend be blurry and less semantically meaningful. Empirically, we observe that performance of RED-Diff improves as $\sigma_y$ decreases. In the noiseless case, RED-Diff achieves higher PSNR and SSIM, but performs worse than OT-ODE in terms of FID and LPIPS (Refer to Figure 12 and 13). Both ΠGDM and OT-ODE achieve similar performance on noiseless inpainting. We further note that noiseless inpainting for OT-ODE can be improved by incorporating null-space decomposition (Wang et al., 2022), which results in improved performance across all datasets. We describe this adjustment in Appendix C.1.

## 5 RELATED WORK

The challenge of solving noisy linear inverse problems without any training has been tackled in many ways, often with other solution concepts than posterior sampling (Elad et al., 2023). Utilizing a diffusion model has a host of recent research that we build upon. Our state-of-the-art baselines ΠGDM (Song et al., 2022) and RED-Diff (Mardani et al., 2023) correspond to lines of research in gradient-based corrections and variational inference.

Earlier gradient-based corrections that approximate $\nabla_{\boldsymbol{x}_t} \ln q(\boldsymbol{y}|\boldsymbol{x}_t)$ in various ways include Diffusion Posterior Sampling (DPS) (Chung et al., 2022a), Manifold Constrained Gradient (Chung et al., 2022b), and an annealed approximation (Jalal et al., 2021). ΠGDM out-performs earlier methods combining adaptive weights and Gaussian posterior approximation with discrete-time denoising diffusion implicit

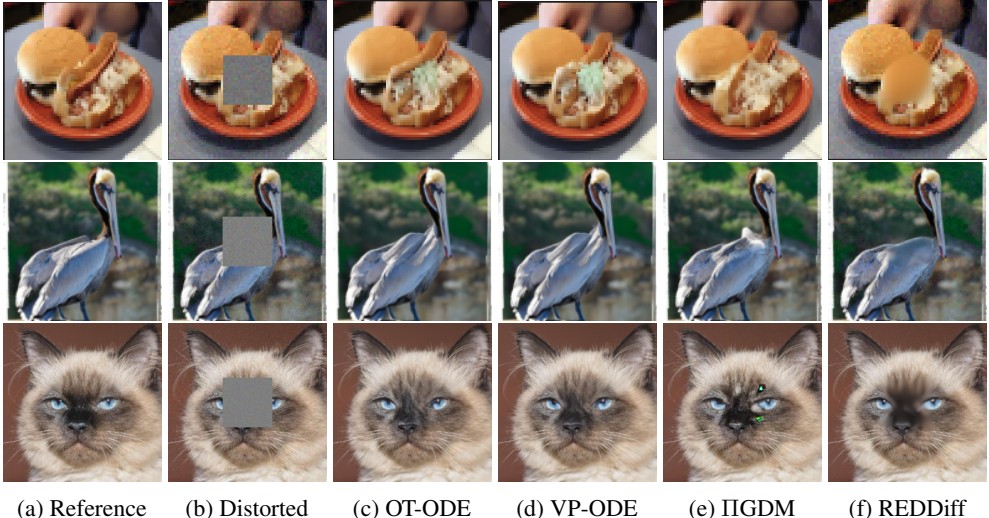

|          |          |          |          |          |          |
|----------|----------|----------|----------|----------|----------|
| (a) Reference | (b) Distorted | (c) OT-ODE | (d) VP-ODE | (e) ΠGDM | (f) REDDiff |

Figure 5: Results for Inpainting (centered mask) with VP-SDE model and $\sigma_y = 0.05$ for (**first row**) ImageNet-64, (**second row**) ImageNet-128, and (**third row**) AFHQ.

model (DDIM) sampling (Song et al., 2021a). Here we adapt ΠGDM to all Gaussian probability paths and to flow sampling. Our results show adaptive weights are unnecessary for strongly performing conditional OT flow sampling. Denoising Diffusion Null Models (DDNM) (Wang et al., 2022) proposed an alternative approximation of $\mathbb{E}_q[\boldsymbol{x}_1|\boldsymbol{x}_t, \boldsymbol{y}]$ using a null-space decomposition specific to linear inverse problems, which has been explored in combination with our method in Appendix C.1.

RED-Diff (Mardani et al., 2023) approximates intractable $q(\boldsymbol{x}_1|\boldsymbol{y})$ directly using variational inference, solving for parameters via optimization. RED-Diff was reported to have mode-seeking behavior confirmed by our results where RED-Diff performed better for noiseless inference. Another earlier variational inference method is Denoising Diffusion Restoration Models (DDRM) (Kawar et al., 2022). DDRM showed SVD can be memory-efficient for image applications, and we adapt their SVD implementations for super-resolution and blur. DDRM incorporates noiseless method ILVR (Choi et al., 2021), and leverages a measurement-dependent forward process (i.e. $q(\boldsymbol{x}_t|\boldsymbol{y}, \boldsymbol{x}_1) \neq q(\boldsymbol{x}_t|\boldsymbol{x}_1)$) like earlier SNIPS (Kawar et al., 2021). SNIPS collapses in special cases to variants proposed in Song & Ermon (2019); Song et al. (2021c); Kadkhodaie & Simoncelli (2020) for linear inverse problems.

## 6 DISCUSSION, LIMITATIONS, AND FUTURE WORK

We have presented a training-free approach to solve linear inverse problems using flows that can leverage either pretrained diffusion or flow models. The algorithm is simple, stable, and requires no hyperparameter tuning when used with conditional OT probability paths. Our method combines past ideas from diffusion including ΠGDM and early starting with the conditional OT probability path from flows, and our results demonstrate that this combination can solve inverse problems for both noisy and noiseless cases across a variety of datasets. Our algorithm using the conditional OT path (OT-ODE) produced results superior to the VP path (VP-ODE) and also to ΠGDM and REDDiff for noisy inverse problems. For the noiseless case, the perceptual quality from OT-ODE is on par with ΠGDM.

One important limitation, shared with most of the past related research, is a restriction to linear observations with scalar variance. Our method can extend to arbitrary covariance, but non-linear observations are more complex. Non-linear observations occur with image inverse tasks when utilizing latent, not pixel-space, diffusion or flow models. Applying our approach to such measurements requires devising an alternative $q^{app}(\boldsymbol{y}|\boldsymbol{x}_t)$. Another shared limitation is that we consider the non-blind setting with known $\boldsymbol{A}$ and $\sigma_y$.

Future research could tackle these limitations. For non-linear observations in latent space, we could perhaps build upon Rout et al. (2023) that uses a latent diffusion model for linear inverses. For the blind setting, we might start from blind extensions to DPS and DDRM (Chung et al., 2023; Murata et al., 2023). As demonstrated here, we may be able to adapt and possibly improve these approaches via conversion to flow sampling using conditional OT paths.

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

## A PROOFS

For clarity, we restate Theorems and Lemmas from the main text before giving their proof.

**Lemma 1.** *For Gaussian probability path $q$ given by Eq. 5, the optimal solution for $\widehat{\boldsymbol{v}}(\boldsymbol{x}_t, \boldsymbol{y})$ is known given $\mathbb{E}_q[\boldsymbol{x}_1|\boldsymbol{x}_t, \boldsymbol{y}]$, and vice versa.*

*Proof.* Inserting Gaussian probability paths defined by Eq. 5 into Eq. 4 gives

$$\mathbb{E}_q\left[\frac{d\boldsymbol{x}_t}{dt}\bigg|\boldsymbol{x}_t, \boldsymbol{y}, \boldsymbol{x}_1\right] = \frac{d\alpha_t}{dt}\boldsymbol{x}_1 + \frac{d\sigma_t}{dt}\left(\frac{\boldsymbol{x}_t - \alpha_t\boldsymbol{x}_1}{\sigma_t}\right). \tag{15}$$

So we reparameterize vector field $\widehat{\boldsymbol{v}}(\boldsymbol{x}_t, \boldsymbol{y}) = \frac{d\alpha_t}{dt}\widehat{\boldsymbol{x}_1}(\boldsymbol{x}_t, \boldsymbol{y}) + \frac{d\sigma_t}{dt}\left(\frac{\boldsymbol{x}_t - \alpha_t\widehat{\boldsymbol{x}_1}(\boldsymbol{x}_t, \boldsymbol{y})}{\sigma_t}\right)$. Inserting these expressions into the Conditional Flow Matching loss gives

$$\int_0^1 \left(\alpha_t\frac{d\ln(\alpha_t/\sigma_t)}{dt}\right)^2 \mathbb{E}_q[(\widehat{\boldsymbol{x}_1} - \boldsymbol{x}_1)^2]dt, \tag{16}$$

recovering the denoising loss with particular $w(t)$. The optimal solution does not depend on the weights and is $\mathbb{E}_q[\boldsymbol{x}_1|\boldsymbol{x}_t, \boldsymbol{y}]$. □

**Lemma 2.** *Consider two Gaussian probability paths $q$ and $q'$ defined by Eq. 5 with $\alpha_t$, $\sigma_t$ and $\alpha'_t$, $\sigma'_t$ respectively. Define $t'(t)$ as the unique solution to $\frac{\sigma_t}{\alpha_t} = \frac{\sigma'_{t'}}{\alpha'_{t'}}$ when it exists for given $t$. Then*

$$\mathbb{E}_q[\boldsymbol{x}_1|\boldsymbol{x}_t, \boldsymbol{y}] = \mathbb{E}_{q'}[\boldsymbol{x}_1|\boldsymbol{X}'_{t'(t)} = \alpha'_{t'(t)}\boldsymbol{x}_t/\alpha_t, \boldsymbol{y}]. \tag{9}$$

*Proof.* We first note that $q$ and $q'$ share $q(\boldsymbol{x}_1|\boldsymbol{y}) = q'(\boldsymbol{x}_1|\boldsymbol{y})$. Then algebraically for Gaussian distributions $q$ and $q'$, $q(\boldsymbol{X}_t = \boldsymbol{x}_t|\boldsymbol{x}_1, \boldsymbol{y}) = q'(\boldsymbol{X}'_{t'(t)} = \alpha'_{t'(t)}\boldsymbol{x}_t/\alpha_t|\boldsymbol{x}_1, \boldsymbol{y})$ when $t'(t)$ exists. The solution for $t'(t)$ is unique due to the monotonicity requirements of both $\alpha$ and $\sigma$. Since the joint densities are therefore identical, $\mathbb{E}_q[\boldsymbol{x}_1|\boldsymbol{x}_t, \boldsymbol{y}] = \mathbb{E}_{q'}[\boldsymbol{x}_1|\boldsymbol{X}'_{t'(t)} = \alpha'_{t'(t)}\boldsymbol{x}_t/\alpha_t,, \boldsymbol{y}]$. □

**Theorem 1.** *Let $q$ be a Gaussian probability path described by Eq. 5. Assume we observe $\boldsymbol{y} \sim q(\boldsymbol{y}|\boldsymbol{x}_1)$ for arbitrary $q(\boldsymbol{y}|\boldsymbol{x}_1)$ and $v(\boldsymbol{x}_t)$ is a vector field enabling sampling $\boldsymbol{x}_t \sim q(\boldsymbol{x}_t)$. Then a vector field $v(\boldsymbol{x}_t, \boldsymbol{y})$ enabling sampling $\boldsymbol{x}_t \sim q(\boldsymbol{x}_t|\boldsymbol{y})$ can be written*

$$v(\boldsymbol{x}_t, \boldsymbol{y}) = v(\boldsymbol{x}_t) + \sigma_t^2 \frac{d\ln(\alpha_t/\sigma_t)}{dt} \nabla_{\boldsymbol{x}_t} \ln q(\boldsymbol{y}|\boldsymbol{x}_t). \tag{11}$$

*Proof.* The optimal $v(\boldsymbol{x}_t, \boldsymbol{y})$ for the Conditional Flow Matching loss is $\mathbb{E}_q[\frac{d\boldsymbol{x}_t}{dt}|\boldsymbol{x}_t, \boldsymbol{y}]$ and for Gaussian probability paths described by Eq. 5 can be written using Eq. 15 as

$$v(\boldsymbol{x}_t, \boldsymbol{y}) = \frac{d\alpha_t}{dt} \mathbb{E}_q[\boldsymbol{x}_1|\boldsymbol{x}_t, \boldsymbol{y}] + \frac{d\sigma_t}{dt} \left( \frac{\boldsymbol{x}_t - \alpha_t \mathbb{E}_q[\boldsymbol{x}_1|\boldsymbol{x}_t, \boldsymbol{y}]}{\sigma_t} \right), \tag{17}$$

where the identical expression without $\boldsymbol{y}$ holds for $v(\boldsymbol{x}_t)$. Inserting the result from Eq. 6 and simplifying gives Eq. 11. □

## B  OUR METHOD FOR ANY GAUSSIAN PROBABILITY PATH

Algorithm 1 in the main text is specific to conditional OT probability paths. Here we provide Algorithm 2 for any Gaussian probability path specified by Eq. 5. Algorithm 1 and Algorithm 2 are written assuming a denoiser $\widehat{\boldsymbol{x}_1}(\boldsymbol{x}_t)$ is provided from a pretrained diffusion model. For completeness, we also include equivalent Algorithm 3 that assumes $\widehat{\boldsymbol{v}}(\boldsymbol{x}_t)$ is provided from a pretrained flow model. In all cases, the vector field or denoiser is evaluated only once per iteration.

Our VP-ODE sampling results correspond to $\alpha_t$ and $\sigma_t$ given from the Variance-Preserving path, which can be found in (Lipman et al., 2022).

---

**Algorithm 2** A training-free approach to solve inverse problems via flows with a pretrained denoiser

---

**Require:** Pretrained denoiser $\widehat{\boldsymbol{x}_1}(\boldsymbol{x}_t)$ converted to Gaussian probability path with $\alpha_t$ and $\sigma_t$ using Section 3.1, noisy measurement $\boldsymbol{y}$, measurement matrix $\boldsymbol{A}$, initial time $t$, adaptive weights $\gamma_t$, and std $\sigma_y$

1: Initialize $\boldsymbol{x}_t = \alpha_t \boldsymbol{y} + \sigma_t \epsilon$, where $\epsilon \sim \mathcal{N}(0, \boldsymbol{I})$  ▷ Initialize $\boldsymbol{x}_t$, Eq. (14)
2: $\boldsymbol{z}_t = \boldsymbol{x}_t$
3: **for** each time step $t'$ of ODE integration **do**  ▷ Integrate ODE from $t' = t$ to 1.
4:     $r_{t'}^2 = \frac{\sigma_{t'}^2}{\sigma_{t'}^2 + \alpha_{t'}^2}$  ▷ Value of $r_t^2$ from Eq. (13)
5:     $\widehat{\boldsymbol{v}} = \left( \alpha_t \frac{d\ln(\alpha_t/\sigma_t)}{dt} \right) \widehat{\boldsymbol{x}_1} + \frac{d\ln\sigma_t}{dt} \boldsymbol{z}_t$  ▷ Convert $\widehat{\boldsymbol{x}_1}$ to vector field, Eq. (8)
6:     $\boldsymbol{g} = (\boldsymbol{y} - \boldsymbol{A}\widehat{\boldsymbol{x}_1})^\top (r_{t'}^2 \boldsymbol{A}\boldsymbol{A}^\top + \sigma_y^2 \boldsymbol{I})^{-1} \boldsymbol{A} \frac{\partial\widehat{\boldsymbol{x}_1}}{\partial\boldsymbol{z}_{t'}}$  ▷ ΠGDM correction
7:     $\widehat{\boldsymbol{v}}_{\text{corrected}} = \widehat{\boldsymbol{v}} + \sigma_t^2 \frac{d\ln(\alpha_t/\sigma_t)}{dt} \gamma_t \boldsymbol{g}$  ▷ Correct unconditional vector field $\widehat{\boldsymbol{v}}$, Eq. (12)
8: **end for**

---

## C  ABLATION STUDY

**Choice of initialization.** We initialize the flow at time $t > 0$ as $\boldsymbol{x}_t = \alpha_t \boldsymbol{y} + \sigma_t \epsilon$ (y-init) where $\epsilon \sim \mathcal{N}(0, \boldsymbol{I})$. Another choice of initialization is to use $\boldsymbol{x}_t = \alpha_t \boldsymbol{A}^\dagger \boldsymbol{y} + \sigma_t \epsilon$. However, empirically we find that this initialization performs worse that y-init on cond-OT model with OT-ODE sampling. We summarize the results of our ablation study in Table 1. We find that on Gaussian deblurring, initialization with $\boldsymbol{A}^\dagger \boldsymbol{y}$ does worse than y-init, while the performance of both the initializations is comparable for super-resolution. In all our experiments, we use y-init, due to its better performance on Gaussian deblurring.

**Ablation over $\gamma_t$ for VP-ODE sampling.** We compare the performance of $\gamma_t = 1$ against $\gamma_t = \sqrt{\frac{\alpha_t}{\alpha_t^2 + \sigma_t^2}}$. We show results of VP-ODE sampling with VP-SDE model in Table 2 and Table 3. As seen our choice of $\gamma_t$ outperform $\gamma_t = 1$ across all the metrics on face-blurred ImageNet-128.

---

**Algorithm 3** A training-free approach to solve inverse problems via flows with a pretrained vector field

---

**Require:** Pretrained vector field $\widehat{\boldsymbol{v}}(\boldsymbol{x}_t)$ converted to Gaussian probability path with $\alpha_t$ and $\sigma_t$ using Section 3.1, noisy measurement $\boldsymbol{y}$, measurement matrix $\boldsymbol{A}$, initial time $t$, adaptive weights $\gamma_t$, and std $\sigma_y$

1: Initialize $\boldsymbol{x}_t = \alpha_t \mathbf{y} + \sigma_t \epsilon$, where $\epsilon \sim \mathcal{N}(0, \boldsymbol{I})$      $\triangleright$ Initialize $\boldsymbol{x}_t$, Eq. (14)

2:   $\boldsymbol{z}_t = \boldsymbol{x}_t$

3: **for** each time step $t'$ of ODE integration **do**      $\triangleright$ Integrate ODE from $t' = t$ to 1.

4:      $\widehat{\boldsymbol{v}} = \widehat{\boldsymbol{v}}(\boldsymbol{z}_{t'})$      $\triangleright$ $\boldsymbol{z}_{t'}$ is value of $\boldsymbol{x}_t$ at time $t'$ during ODE integration

5:      $r_{t'}^2 = \frac{\sigma_{t'}^2}{\sigma_{t'}^2 + \alpha_{t'}^2}$      $\triangleright$ Value of $r_t^2$ from Eq. (13)

6:      $\widehat{\boldsymbol{x}_1} = \left( \alpha_t \frac{d \ln(\alpha_t/\sigma_t)}{dt} \right)^{-1} \left( \widehat{\boldsymbol{v}} - \frac{d \ln \sigma_t}{dt} \boldsymbol{z}_t \right)$      $\triangleright$ Convert vector field to $\widehat{\boldsymbol{x}_1}$, Eq. (8)

7:      $\boldsymbol{g} = (\boldsymbol{y} - \boldsymbol{A}\widehat{\boldsymbol{x}_1})^\top (r_{t'}^2 \boldsymbol{A}\boldsymbol{A}^\top + \sigma_y^2 \boldsymbol{I})^{-1} \boldsymbol{A} \frac{\partial \widehat{\boldsymbol{x}_1}}{\partial \boldsymbol{z}_{t'}}$      $\triangleright$ ΠGDM correction

8:      $\widehat{\boldsymbol{v}}_{\text{corrected}} = \widehat{\boldsymbol{v}} + \sigma_t^2 \frac{d \ln(\alpha_t/\sigma_t)}{dt} \gamma_t \boldsymbol{g}$      $\triangleright$ Correct unconditional vector field $\widehat{\boldsymbol{v}}$, Eq. (12)

9: **end for**

---

Table 1: Quantitative evaluation of choice of initialization for conditional OT flow model with OT-ODE sampling on AFHQ dataset. We find that y-init outperforms $\boldsymbol{A}^\dagger \boldsymbol{y}$ on Gaussian deblurring.

| Initialization | Start time | NFEs ↓ | Gaussian deblur, $\sigma_y = 0.05$ | | | | SR 4×, $\sigma_y = 0.05$ | | | |
|---|---|---|---|---|---|---|---|---|---|---|
| | | | FID ↓ | LPIPS ↓ | PSNR ↑ | SSIM ↑ | FID ↓ | LPIPS ↓ | PSNR ↑ | SSIM ↑ |
| y init | 0.2 | 100 | **7.57** | **0.268** | **30.28** | **0.626** | **6.03** | **0.219** | **31.12** | **0.739** |
| $\boldsymbol{A}^\dagger \boldsymbol{y}$ | 0.1 | 100 | 41.22 | 0.449 | 28.79 | 0.392 | 12.93 | 0.292 | 30.46 | 0.664 |
| $\boldsymbol{A}^\dagger \boldsymbol{y}$ | 0.2 | 100 | 56.42 | 0.554 | 28.11 | 0.249 | 6.09 | 0.219 | 31.12 | 0.739 |

Table 2: Quantitative evaluation of value of $\gamma_t$ in VP-ODE sampling with VP-SDE model on face-blurred ImageNet-128 dataset.

| $\gamma_t$ | Start time | NFEs ↓ | SR 2×, $\sigma_y = 0.05$ | | | | Gaussian deblur, $\sigma_y = 0.05$ | | | |
|---|---|---|---|---|---|---|---|---|---|---|
| | | | FID ↓ | LPIPS ↓ | PSNR ↑ | SSIM ↑ | FID ↓ | LPIPS ↓ | PSNR ↑ | SSIM ↑ |
| $1$ | 0.4 | 60 | 32.66 | 0.371 | 29.06 | 0.530 | 29.31 | 0.346 | 29.12 | 0.554 |
| $\sqrt{\frac{\alpha_t}{\alpha_t^2 + \sigma_t^2}}$ | 0.4 | 60 | **9.14** | **0.167** | **32.06** | **0.838** | **10.14** | **0.196** | **31.59** | **0.800** |

Table 3: Quantitative evaluation of value of $\gamma_t$ in VP-ODE sampling with VP-SDE model on face-blurred ImageNet-128 dataset.

| $\gamma_t$ | Start time | NFEs ↓ | Inpainting-*Center*, $\sigma_y = 0.05$ | | | | Denoising, $\sigma_y = 0.05$ | | | |
|---|---|---|---|---|---|---|---|---|---|---|
| | | | FID ↓ | LPIPS ↓ | PSNR ↑ | SSIM ↑ | FID ↓ | LPIPS ↓ | PSNR ↑ | SSIM ↑ |
| $1$ | 0.3 | 70 | 53.03 | 0.285 | 31.55 | 0.737 | 28.37 | 0.238 | 31.63 | 0.786 |
| $\sqrt{\frac{\alpha_t}{\alpha_t^2 + \sigma_t^2}}$ | 0.3 | 70 | **8.47** | **0.129** | **34.43** | **0.876** | **5.83** | **0.087** | **35.85** | **0.938** |

**Variation of performance with NFEs.** We analyze the variation in performance of OT-ODE, VP-ODE and ΠGDM for solving linear inverse problems as NFEs are varied. The results have been summarized in Figure 6. We observe that OT-ODE consistently outperforms VP-ODE and ΠGDM across all measurements in terms of FID and LPIPS metrics, even for NFEs as small as 20. We also note that the choice of starting time matters to achieve good performance with OT-ODE. For instance, starting at $t = 0.4$ outperforms $t = 0.2$ when NFEs are small, but eventually as NFEs is increased, $t = 0.2$ performs better. We also note that ΠGDM achieves higher values of PSNR and SSIM at smaller NFEs for super-resolution but has inferior FID and LPIPS compared to OT-ODE.

**Choice of starting time.** We plot the variation in performance of OT-ODE and VP-ODE sampling with change in start times for conditional OT model and VP-SDE model on AFHQ dataset in Figure 7 and Figure 8, respectively. We note that in general, OT-ODE sampling achieves optimal performance across all measurements and all metrics at $t = 0.2$ while VP-ODE sampling achieves optimal performance between start times of $t = 0.3$ and $0.4$. In this work, for all the experiments, we use $t = 0.2$ for OT-ODE sampling and $t = 0.4$ for VP-ODE sampling.

## C.1 Noiseless Null and Range Space Decomposition

When $\sigma_y^2 = 0$, we can produce a vector field approximation with even lower Conditional Flow Matching loss by applying a null-space and range-space decomposition motivated by DDNM (Wang et al., 2022). In particular, when $\boldsymbol{y} = \boldsymbol{A}\boldsymbol{x}_1$, we have that $\boldsymbol{A}^\dagger \boldsymbol{y} = \boldsymbol{A}^\dagger \boldsymbol{A}\boldsymbol{x}_1$ (where $\boldsymbol{A}^\dagger$ is the pseudo-inverse of $\boldsymbol{A}$) and so

$$\mathbb{E}_q[\boldsymbol{x}_1|\boldsymbol{x}_t, \boldsymbol{y}] = \mathbb{E}_q[\boldsymbol{A}^\dagger \boldsymbol{A}\boldsymbol{x}_1 + (\boldsymbol{I} - \boldsymbol{A}^\dagger \boldsymbol{A})\boldsymbol{x}_1|\boldsymbol{x}_t, \boldsymbol{y}] = \boldsymbol{A}^\dagger \boldsymbol{y} + (\boldsymbol{I} - \boldsymbol{A}^\dagger \boldsymbol{A})\mathbb{E}_q[\boldsymbol{x}_1|\boldsymbol{x}_t, \boldsymbol{y}]. \quad (18)$$

So when $\sigma_y^2 = 0$, it is only necessary to approximate the second term, as the first term is known through $\boldsymbol{y}$. The regression loss is minimized for the first term automatically and $\widehat{\boldsymbol{x}_1}(\boldsymbol{x}_t, \boldsymbol{y})$ is only responsible for predicting the second term.

In our experiments, we find that null space decomposition helps in inpainting but not other measurements. We summarize the results in Table 4 to 9.

Table 4: Comparison of performance OT-ODE sampling and OT-ODE sampling with null and range space decomposition (NRSD) on face-blurred ImageNet-$64 \times 64$. For inpainting, OT-ODE sampling with null and range space decomposition outperforms simple OT-ODE sampling.

| Model | Inference | NFEs ↓ | Inpainting-*Center*, $\sigma_y = 0$ | | | |
|---|---|---|---|---|---|---|
| | | | FID ↓ | LPIPS ↓ | PSNR ↑ | SSIM ↑ |
| OT | OT-ODE | 80 | 4.94 | 0.080 | 37.42 | 0.885 |
| OT | OT-ODE-NRSD | 80 | **3.84** | **0.072** | **38.23** | **0.888** |
| OT | VP-ODE | 80 | 7.85 | 0.120 | 34.24 | 0.858 |
| VP-SDE | OT-ODE | 80 | 4.85 | 0.079 | 37.64 | 0.887 |
| VP-SDE | OT-ODE-NRSD | 80 | **3.77** | **0.072** | **38.24** | **0.888** |
| VP-SDE | VP-ODE | 80 | 7.21 | 0.117 | 34.33 | 0.860 |

Table 5: Comparison of performance OT-ODE sampling and OT-ODE sampling with null and range space decomposition (NRSD) on face-blurred ImageNet-$64 \times 64$. For tasks like super-resolution and Gaussian deblurring, OT-ODE sampling without null and range space decomposition outperforms other methods.

| Model | Inference | NFEs ↓ | SR 2×, $\sigma_y = 0$ | | | | Gaussian deblur, $\sigma_y = 0$ | | | |
|---|---|---|---|---|---|---|---|---|---|---|
| | | | FID ↓ | LPIPS ↓ | PSNR ↑ | SSIM ↑ | FID ↓ | LPIPS ↓ | PSNR ↑ | SSIM ↑ |
| OT | OT-ODE | 80 | **6.46** | **0.119** | **31.59** | **0.839** | **2.59** | **0.038** | **35.31** | **0.961** |
| OT | OT-ODE-NRSD | 80 | 7.37 | 0.134 | 31.05 | 0.799 | 3.05 | 0.044 | 35.19 | 0.956 |
| OT | VP-ODE | 80 | 8.29 | 0.147 | 31.20 | 0.817 | 6.13 | 0.083 | 33.31 | 0.929 |
| VP-SDE | OT-ODE | 80 | **6.32** | **0.118** | **31.60** | **0.839** | **2.61** | **0.037** | **35.45** | **0.963** |
| VP-SDE | OT-ODE-NRSD | 80 | 7.13 | 0.133 | 31.06 | 0.798 | 2.99 | 0.044 | 35.24 | 0.956 |
| VP-SDE | VP-ODE | 80 | 7.76 | 0.145 | 31.21 | 0.817 | 5.68 | 0.080 | 33.37 | 0.931 |

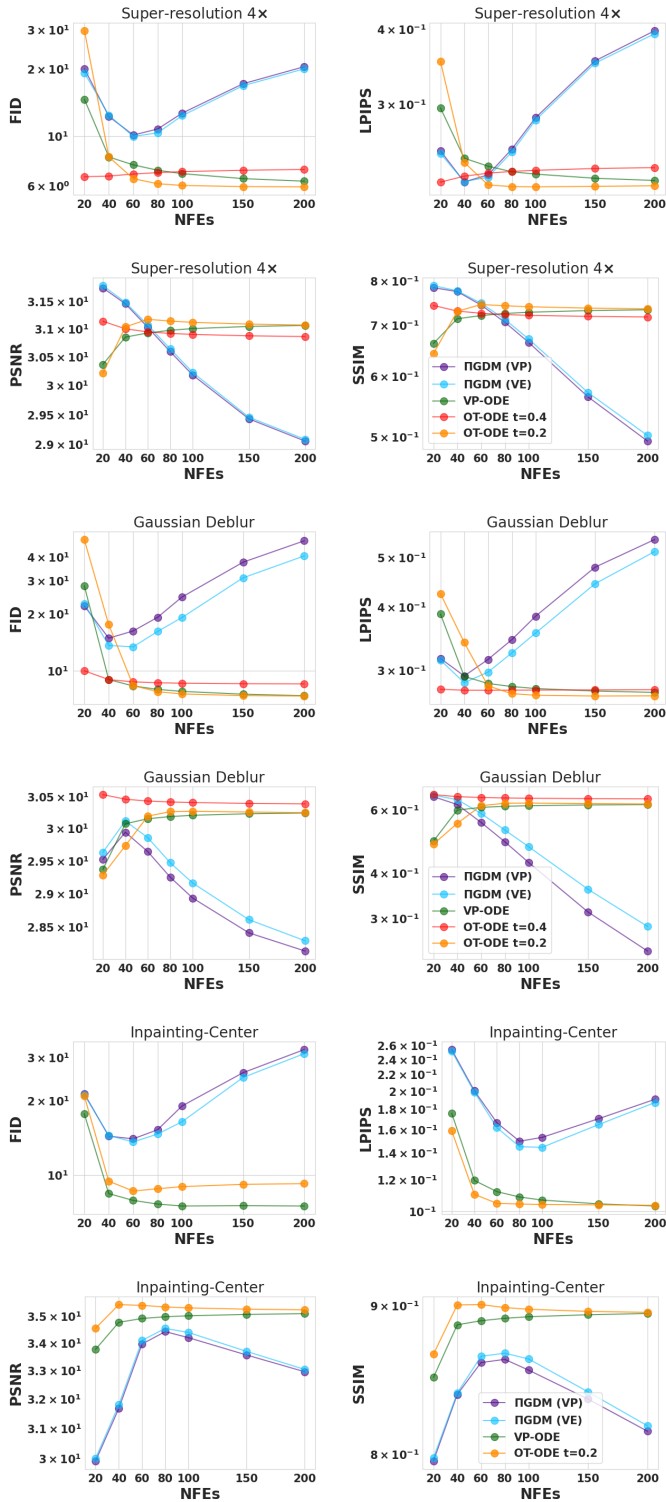

Figure 6: Performance of different procedures for solving linear inverse problems with variation in NFEs on AFHQ dataset. We use pretrained conditional OT model and set $\sigma_y = 0.05$. The legends VP and VE indicate the choice of $r_t^2$ used in ΠGDM (See Appendix E.1). Time $t = 0.2$ and $0.4$ indicates the starting time of sampling with OT-ODE.

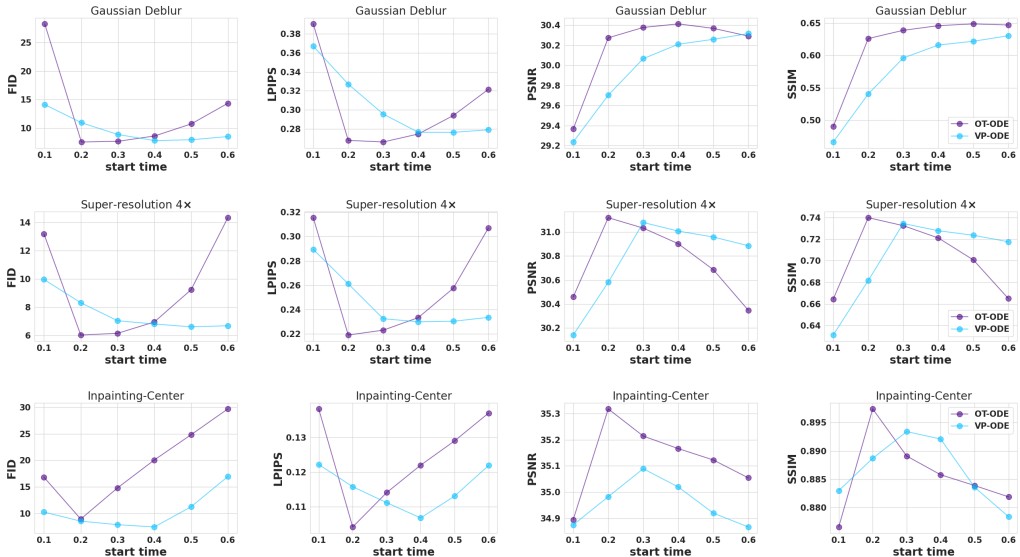

Figure 7: Performance of OT-ODE and VP-ODE in solving linear inverse problems with varying start times on AFHQ dataset. We use pretrained cond-OT model and set $\sigma_y = 0.05$.

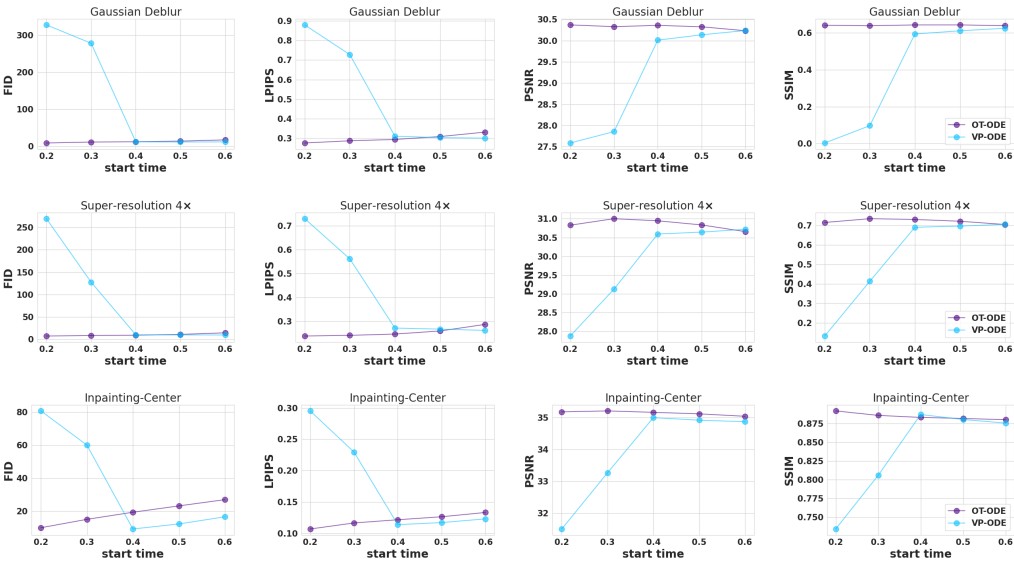

Figure 8: Performance of OT-ODE and VP-ODE in solving linear inverse problems with varying start times on AFHQ dataset. We use pretrained VP-SDE model and set $\sigma_y = 0.05$.

# D ADDITIONAL EMPIRICAL RESULTS

## D.1 FIGURES FOR CONDITIONAL OT FLOW MATCHING MODEL

The main text includes figures produced with the denoiser from the continuous-time VP-SDE diffusion model. Here we provide the same figures using our pre-trained conditional OT flow matching model using Algorithm 3 instead. To save compute, here we only include our ΠGDM baseline as RED-Diff required extensive hyperparameter tuning. The qualitative and quantitative results using the flow model instead of diffusion model checkpoint are identical, where we observe the best performance with OT-ODE inference. We provide plots of various metrics across all datasets and tasks for the cond-OT model in Figure 14 for $\sigma_y = 0.05$ and Figure 13 for $\sigma_y = 0$, respectively.

Table 6: Comparison of performance OT-ODE sampling and OT-ODE sampling with null and range space decomposition (NRSD) on face-blurred ImageNet-$128 \times 128$.

| Model | Inference | NFEs ↓ | SR 2×, $\sigma_y = 0$ | | | | Gaussian deblur, $\sigma_y = 0$ | | | |
|---|---|---|---|---|---|---|---|---|---|---|
| | | | FID ↓ | LPIPS ↓ | PSNR ↑ | SSIM ↑ | FID ↓ | LPIPS ↓ | PSNR ↑ | SSIM ↑ |
| OT | OT-ODE | 70 | 4.46 | **0.097** | **33.88** | **0.903** | 2.09 | 0.048 | 37.49 | 0.961 |
| OT | OT-ODE-NRSD | 70 | **3.62** | 0.099 | 33.24 | 0.876 | **1.42** | **0.036** | **38.35** | **0.969** |
| OT | VP-ODE | 70 | 7.69 | 0.144 | 32.93 | 0.871 | 6.02 | 0.108 | 34.73 | 0.925 |
| VP-SDE | OT-ODE | 70 | 4.62 | **0.096** | **33.95** | **0.906** | 2.26 | 0.046 | 37.79 | 0.967 |
| VP-SDE | OT-ODE-NRSD | 70 | **3.44** | 0.098 | 33.28 | 0.877 | **1.36** | **0.035** | **38.44** | **0.969** |
| VP-SDE | VP-ODE | 70 | 7.91 | 0.144 | 32.87 | 0.869 | 5.64 | 0.105 | 34.81 | 0.928 |

Table 7: Comparison of performance OT-ODE sampling and OT-ODE sampling with null and range space decomposition (NRSD) on face-blurred ImageNet-$128 \times 128$

| Model | Inference | NFEs ↓ | Inpainting-*Center*, $\sigma_y = 0$ | | | |
|---|---|---|---|---|---|---|
| | | | FID ↓ | LPIPS ↓ | PSNR ↑ | SSIM ↑ |
| OT | OT-ODE | 70 | 5.88 | 0.095 | 37.06 | 0.894 |
| OT | OT-ODE-NRSD | 70 | **3.95** | **0.074** | **38.27** | **0.906** |
| OT | VP-ODE | 70 | 8.63 | 0.144 | 34.48 | 0.864 |
| VP-SDE | OT-ODE | 70 | 5.93 | 0.094 | 37.31 | 0.898 |
| VP-SDE | OT-ODE-NRSD | 70 | **3.84** | **0.073** | **38.27** | **0.906** |
| VP-SDE | VP-ODE | 70 | 8.08 | 0.142 | 34.55 | 0.865 |

Table 8: Comparison of performance OT-ODE sampling and OT-ODE sampling with null and range space decomposition (NRSD) on AFHQ-$256 \times 256$

| Model | Inference | NFEs ↓ | SR 4×, $\sigma_y = 0$ | | | | Gaussian deblur, $\sigma_y = 0$ | | | |
|---|---|---|---|---|---|---|---|---|---|---|
| | | | FID ↓ | LPIPS ↓ | PSNR ↑ | SSIM ↑ | FID ↓ | LPIPS ↓ | PSNR ↑ | SSIM ↑ |
| OT | OT-ODE | 100 | 5.75 | **0.169** | **32.25** | **0.792** | **6.63** | **0.213** | **31.29** | **0.722** |
| OT | OT-ODE-NRSD | 100 | **5.73** | 0.179 | 31.69 | 0.753 | 7.32 | 0.237 | 30.72 | 0.665 |
| OT | VP-ODE | 100 | 6.14 | 0.194 | 31.93 | 0.773 | 7.38 | 0.231 | 31.10 | 0.705 |
| VP-SDE | OT-ODE | 100 | **6.58** | **0.178** | **32.18** | **0.789** | **8.24** | **0.226** | **31.21** | **0.717** |
| VP-SDE | OT-ODE-NRSD | 100 | 6.99 | 0.195 | 31.65 | 0.752 | 10.19 | 0.255 | 30.66 | 0.662 |
| VP-SDE | VP-ODE | 100 | 8.00 | 0.225 | 31.48 | 0.742 | 9.19 | 0.252 | 30.91 | 0.688 |

Table 9: Comparison of performance OT-ODE sampling and OT-ODE sampling with null and range space decomposition (NRSD) on AFHQ-$256 \times 256$

| Model | Inference | NFEs ↓ | Inpainting-*Center*, $\sigma_y = 0$ | | | | Inpainting-*Free-form*, $\sigma_y = 0$ | | | |
|---|---|---|---|---|---|---|---|---|---|---|
| | | | FID ↓ | LPIPS ↓ | PSNR ↑ | SSIM ↑ | FID ↓ | LPIPS ↓ | PSNR ↑ | SSIM ↑ |
| OT | OT-ODE | 100 | 8.87 | 0.061 | 37.45 | **0.921** | 4.98 | 0.097 | 36.15 | 0.889 |
| OT | OT-ODE-NRSD | 100 | **7.95** | **0.046** | **38.01** | **0.921** | **4.12** | **0.083** | **36.62** | **0.890** |
| OT | VP-ODE | 100 | 9.18 | 0.106 | 35.63 | 0.898 | 6.92 | 0.135 | 34.72 | 0.869 |
| VP-SDE | OT-ODE | 100 | **9.95** | 0.064 | 37.49 | **0.918** | 5.39 | 0.099 | 36.15 | **0.887** |
| VP-SDE | OT-ODE-NRSD | 100 | 10.96 | **0.052** | **37.95** | 0.916 | **4.87** | **0.089** | **36.52** | 0.884 |
| VP-SDE | VP-ODE | 100 | 10.50 | 0.112 | 35.59 | 0.893 | 7.36 | 0.139 | 34.65 | 0.865 |

Table 10: Quantitative evaluation of linear inverse problems on face-blurred ImageNet-$64 \times 64$

| Model | Inference | NFEs ↓ | SR 2×, $\sigma_y = 0.05$ | | | | Gaussian deblur, $\sigma_y = 0.05$ | | | |
|---|---|---|---|---|---|---|---|---|---|---|
| | | | FID ↓ | LPIPS ↓ | PSNR ↑ | SSIM ↑ | FID ↓ | LPIPS ↓ | PSNR ↑ | SSIM ↑ |
| OT | OT-ODE | 80 | **6.07** | **0.157** | **30.88** | **0.799** | **6.83** | **0.185** | **30.51** | **0.773** |
| OT | VP-ODE | 80 | 7.82 | 0.163 | 30.75 | 0.792 | 8.72 | 0.190 | 30.40 | 0.765 |
| OT | ΠGDM | 100 | 6.52 | 0.168 | 30.54 | 0.753 | 55.19 | 0.374 | 28.74 | 0.516 |
| VP-SDE | OT-ODE | 80 | **5.57** | **0.155** | 30.88 | 0.799 | **6.33** | **0.181** | **30.52** | 0.773 |
| VP-SDE | VP-ODE | 80 | 7.40 | 0.160 | 30.75 | 0.792 | 8.16 | 0.187 | 30.42 | 0.766 |
| VP-SDE | ΠGDM | 100 | 6.84 | 0.174 | 30.48 | 0.743 | 54.77 | 0.376 | 28.74 | 0.511 |
| VP-SDE | RED-Diff | 1000 | 23.02 | 0.187 | **31.22** | **0.839** | 51.20 | 0.236 | 30.19 | **0.776** |

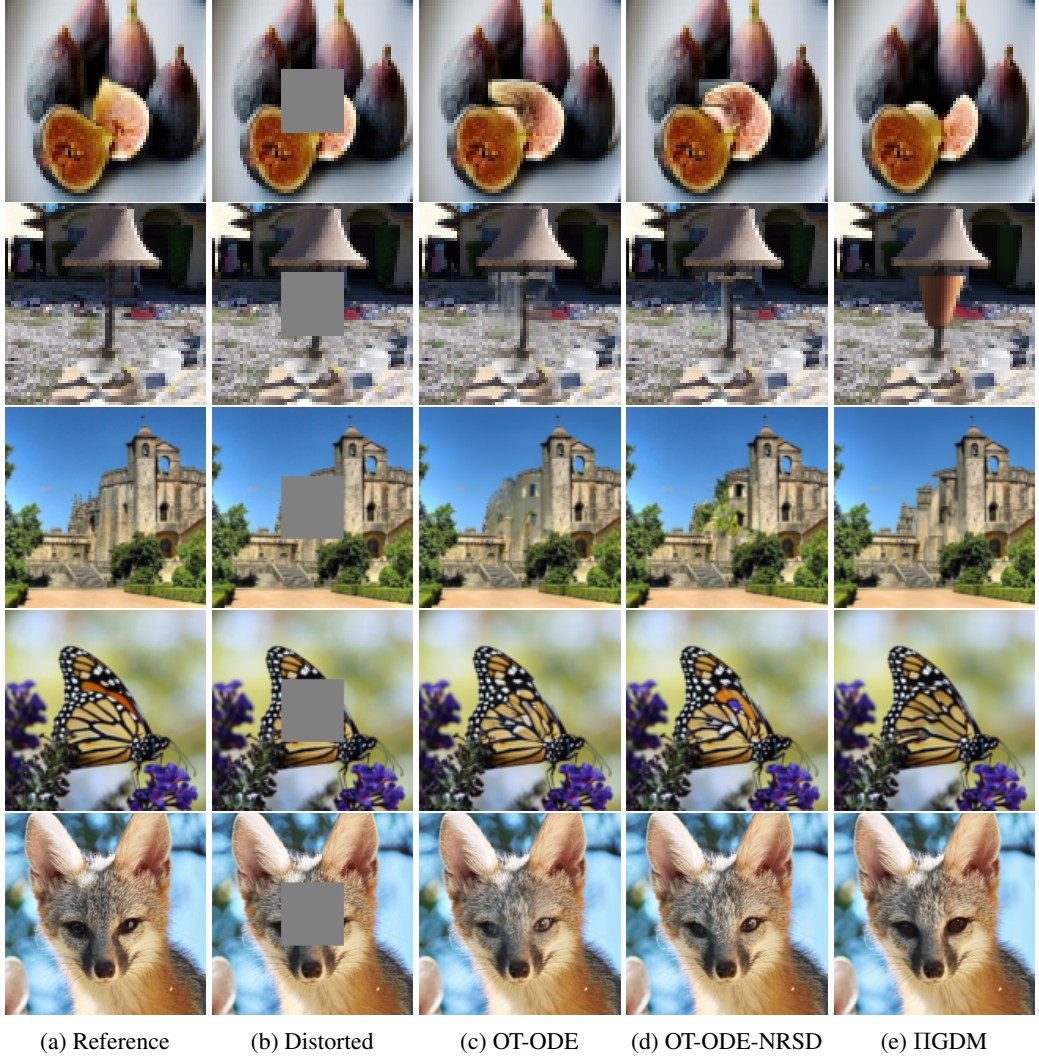

|          (a) Reference          |          (b) Distorted          |          (c) OT-ODE          |          (d) OT-ODE-NRSD          |          (e) ΠGDM          |

Figure 9: Comparison of inpainting (center mask) via OT-ODE sampling with and without null and range space decomposition (NRSD). We use conditional OT model and $\sigma_y = 0$ for (**first and second row**) face-blurred ImageNet-64, (**third row**) face-blurred ImageNet-128, and (**fourth row**) AFHQ.

Table 11: Quantitative evaluation of linear inverse problems on face-blurred ImageNet-$64 \times 64$

| Model | Inference | NFEs ↓ | Inpainting-*Center*, $\sigma_y = 0.05$ | | | | Denoising, $\sigma_y = 0.05$ | | | |
|---|---|---|---|---|---|---|---|---|---|---|
| | | | FID ↓ | LPIPS ↓ | SSIM ↑ | PSNR ↑ | FID ↓ | LPIPS ↓ | SSIM ↑ | PSNR ↑ |
| OT | OT-ODE | 80 | **5.45** | **0.101** | **34.21** | **0.870** | 2.91 | 0.044 | 35.96 | 0.968 |
| OT | VP-ODE | 80 | 5.70 | 0.105 | 33.87 | 0.865 | 3.54 | 0.049 | 35.37 | 0.960 |
| OT | ΠGDM | 100 | 9.25 | 0.111 | 34.13 | 0.863 | 16.59 | 0.102 | 34.60 | 0.906 |
| DDPM | OT-ODE | 80 | **5.03** | **0.098** | **34.25** | **0.872** | 2.76 | 0.042 | 36.02 | 0.969 |
| VP-SDE | VP-ODE | 80 | 5.26 | 0.103 | 33.93 | 0.866 | 3.29 | 0.048 | 35.45 | 0.961 |
| VP-SDE | ΠGDM | 100 | 9.75 | 0.113 | 34.03 | 0.860 | 17.19 | 0.107 | 34.25 | 0.901 |
| VP-SDE | RED-Diff | 1000 | 12.18 | 0.119 | 33.97 | 0.881 | 6.02 | 0.041 | 35.64 | 0.964 |

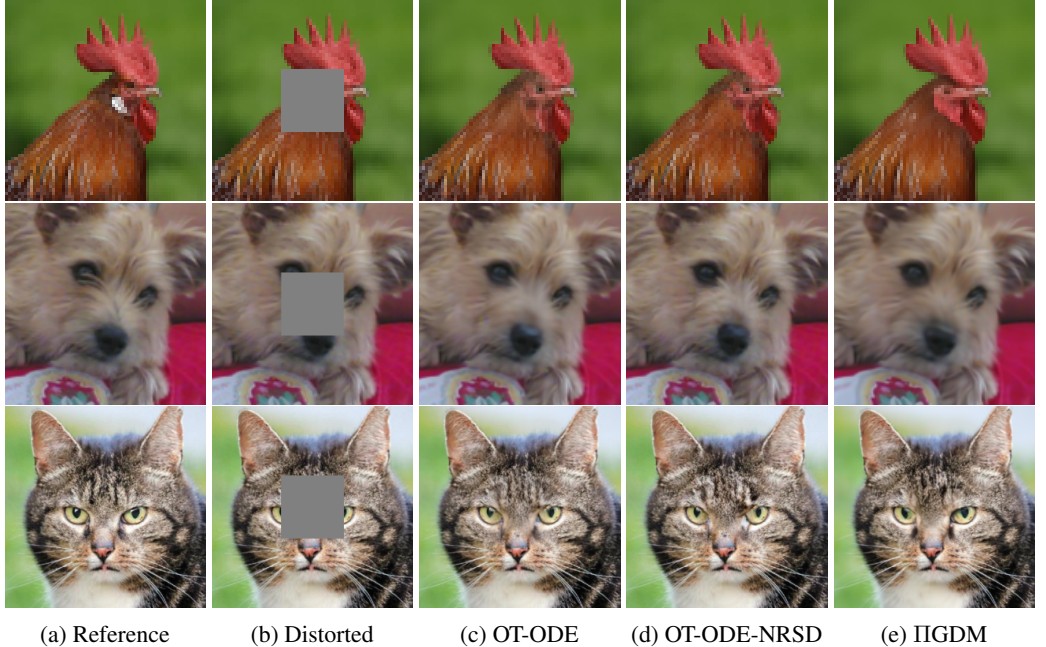

(a) Reference     (b) Distorted     (c) OT-ODE     (d) OT-ODE-NRSD     (e) ΠGDM

Figure 10: Comparison of inpainting (center mask) via OT-ODE sampling with and without null and range space decomposition (NRSD) for (**first row**) face-blurred ImageNet-64, (**second row**) face-blurred ImageNet-128, and (**third row**) AFHQ. We use VP-SDE model and $\sigma_y = 0$.

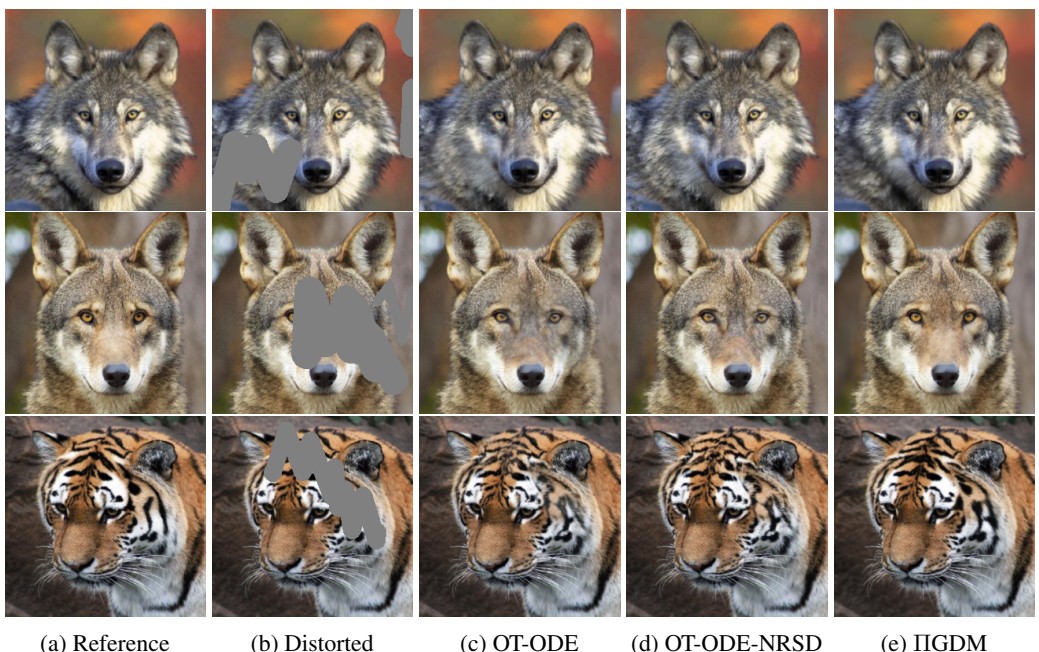

(a) Reference     (b) Distorted     (c) OT-ODE     (d) OT-ODE-NRSD     (e) ΠGDM

Figure 11: Comparison of inpainting (free-form mask) via OT-ODE sampling with and without null and range space decomposition (NRSD) for AFHQ. We use conditional OT model and $\sigma_y = 0$.

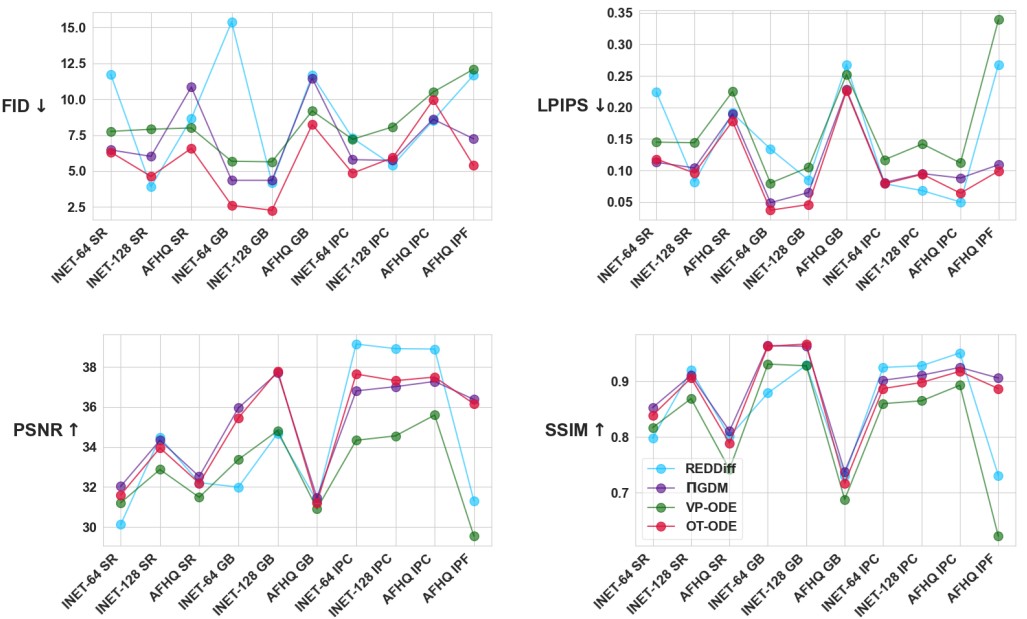

Figure 12: Quantitative evaluation of pretrained VP-SDE model for linear inverse problems on super-resolution (SR), gaussian deblurring (GB), image inpainting - centered mask (IPC) and inpainting - free-form (IPF) with $\sigma_y = 0$. We show results on face-blurred ImageNet-64 (INET-64), face-blurred ImageNet-128 (INET-128), and AFHQ-256 (AFHQ).

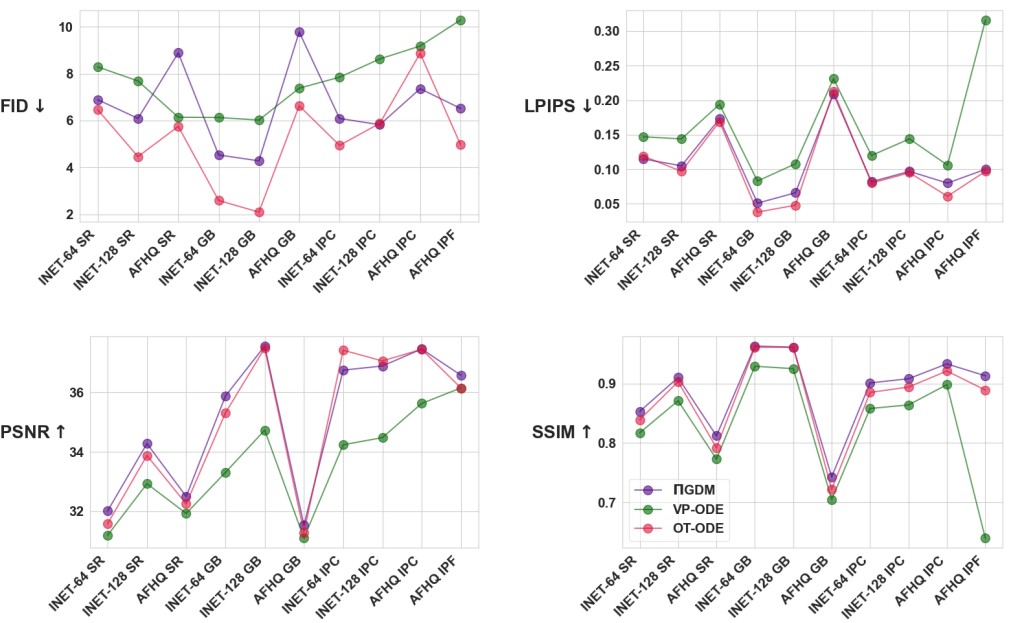

Figure 13: Quantitative evaluation of pretrained conditional OT model for linear inverse problems on super-resolution (SR), gaussian deblurring (GB), image inpainting - centered mask (IPC) and inpainting - freeform (IPF) with $\sigma_y = 0$. We show results on face-blurred ImageNet-64 (INET-64), face-blurred ImageNet-128 (INET-128), and AFHQ-256 (AFHQ).

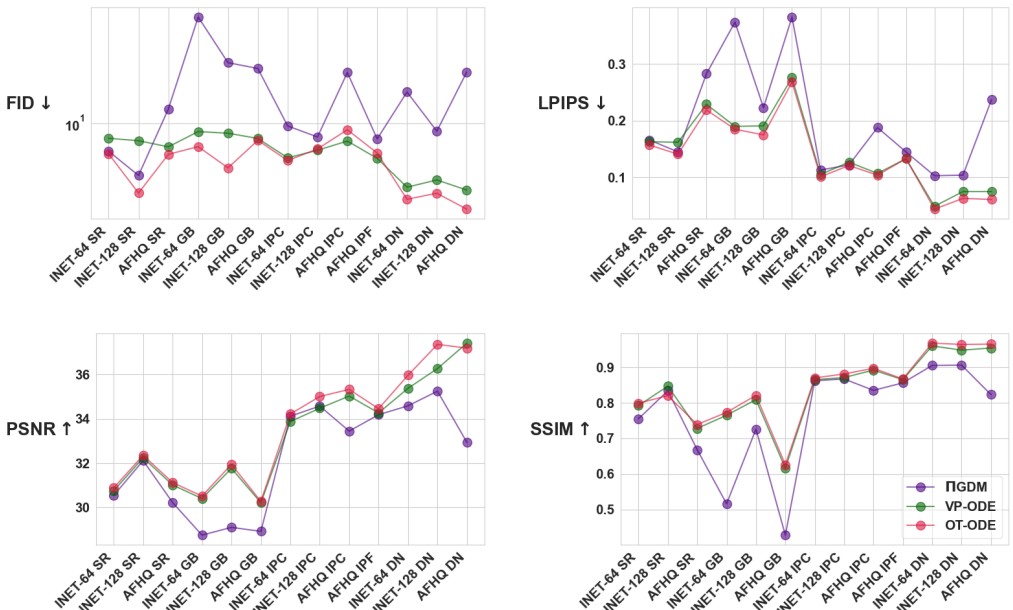

Figure 14: Quantitative evaluation of pretrained conditional OT model for linear inverse problems on super-resolution (SR), gaussian deblurring (GB), image inpainting - centered mask(IPC) and denoising (DN) with $\sigma_y = 0.05$. We show results on face-blurred ImageNet-64 (INET-64), face-blurred ImageNet-128 (INET-128), and AFHQ-256 (AFHQ).

Table 12: Quantitative evaluation of linear inverse problems on face-blurred ImageNet-$64 \times 64$

| Model | Inference | NFEs ↓ | SR 2×, $\sigma_y = 0.01$ | | | | Gaussian deblur, $\sigma_y = 0.01$ | | | |
|---|---|---|---|---|---|---|---|---|---|---|
| | | | FID ↓ | LPIPS ↓ | PSNR ↑ | SSIM ↑ | FID ↓ | LPIPS ↓ | PSNR ↑ | SSIM ↑ |
| OT | OT-ODE | 80 | **5.39** | **0.114** | **31.64** | **0.838** | **4.68** | **0.097** | **32.09** | **0.891** |
| OT | VP-ODE | 80 | 5.90 | 0.120 | 31.45 | 0.832 | 5.58 | 0.103 | 31.90 | 0.884 |
| OT | ΠGDM | 100 | 8.14 | 0.167 | 30.01 | 0.765 | 26.65 | 0.223 | 29.10 | 0.726 |
| VP-SDE | OT-ODE | 80 | **5.11** | **0.113** | 31.64 | 0.838 | **4.49** | **0.094** | **32.11** | **0.892** |
| VP-SDE | VP-ODE | 80 | 5.61 | 0.119 | 31.46 | 0.831 | 5.27 | 0.101 | 31.92 | 0.885 |
| VP-SDE | ΠGDM | 100 | 9.08 | 0.182 | 29.82 | 0.746 | 26.31 | 0.238 | 28.99 | 0.712 |
| VP-SDE | RED-Diff | 1000 | 12.78 | 0.131 | **32.26** | **0.884** | 30.65 | 0.194 | 31.29 | 0.832 |

Table 13: Quantitative evaluation of linear inverse problems on face-blurred ImageNet-$64 \times 64$

| Model | Inference | NFEs ↓ | Inpainting-*Center*, $\sigma_y = 0.01$ | | | | Denoising, $\sigma_y = 0.01$ | | | |
|---|---|---|---|---|---|---|---|---|---|---|
| | | | FID ↓ | LPIPS ↓ | PSNR ↑ | SSIM ↑ | FID ↓ | LPIPS ↓ | PSNR ↑ | SSIM ↑ |
| OT | OT-ODE | 80 | **3.86** | **0.070** | **37.60** | **0.894** | **0.57** | **0.007** | 45.21 | 0.994 |
| OT | VP-ODE | 80 | 4.64 | 0.085 | 36.12 | 0.884 | 1.96 | 0.026 | 39.62 | 0.981 |
| OT | ΠGDM | 100 | 31.14 | 0.233 | 29.73 | 0.744 | 61.19 | 0.298 | 28.88 | 0.703 |
| VP-SDE | OT-ODE | 80 | **3.75** | **0.069** | 37.77 | 0.896 | **0.62** | 0.006 | **46.02** | **0.995** |
| VP-SDE | VP-ODE | 80 | 4.34 | 0.083 | 36.19 | 0.885 | 1.85 | 0.025 | 39.77 | 0.981 |
| VP-SDE | ΠGDM | 100 | 37.87 | 0.284 | 29.32 | 0.671 | 69.57 | 0.382 | 28.47 | 0.573 |
| VP-SDE | RED-Diff | 1000 | 7.52 | 0.083 | **38.32** | **0.921** | 0.63 | **0.004** | 45.89 | **0.995** |

Table 14: Quantitative evaluation of linear inverse problems on face-blurred ImageNet-128 × 128

| Model | Inference | NFEs ↓ | SR 2×, $\sigma_y = 0.05$ | | | | Gaussian deblur, $\sigma_y = 0.05$ | | | |
|---|---|---|---|---|---|---|---|---|---|---|
| | | | FID ↓ | LPIPS ↓ | PSNR ↑ | SSIM ↑ | FID ↓ | LPIPS ↓ | PSNR ↑ | SSIM ↑ |
| OT | OT-ODE | 70 | **3.22** | **0.141** | **32.35** | 0.820 | **4.84** | **0.175** | **31.94** | **0.821** |
| OT | VP-ODE | 70 | 7.52 | 0.162 | 32.24 | **0.847** | 8.49 | 0.191 | 31.76 | 0.809 |
| OT | ΠGDM | 100 | 4.38 | 0.148 | 32.07 | 0.831 | 30.30 | 0.328 | 29.96 | 0.606 |
| VP-SDE | OT-ODE | 70 | **3.21** | **0.139** | **32.40** | **0.855** | **4.49** | **0.173** | **32.02** | **0.824** |
| VP-SDE | VP-ODE | 70 | 9.14 | 0.166 | 32.06 | 0.838 | 9.35 | 0.193 | 31.66 | 0.804 |
| VP-SDE | ΠGDM | 100 | 7.55 | 0.183 | 31.61 | 0.785 | 55.61 | 0.463 | 28.57 | 0.414 |
| VP-SDE | RED-Diff | 1000 | 10.54 | 0.182 | 31.82 | 0.852 | 21.43 | 0.229 | 31.41 | 0.807 |

Table 15: Quantitative evaluation of linear inverse problems on face-blurred ImageNet-128 × 128

| Model | Inference | NFEs ↓ | Inpainting-*Center*, $\sigma_y = 0.05$ | | | | Denoising, $\sigma_y = 0.05$ | | | |
|---|---|---|---|---|---|---|---|---|---|---|
| | | | FID ↓ | LPIPS ↓ | PSNR ↑ | SSIM ↑ | FID ↓ | LPIPS ↓ | PSNR ↑ | SSIM ↑ |
| OT | OT-ODE | 70 | 6.58 | **0.121** | **35.00** | **0.881** | **3.21** | **0.063** | **37.35** | **0.964** |
| OT | VP-ODE | 70 | **6.44** | 0.127 | 34.47 | 0.871 | 3.98 | 0.075 | 36.26 | 0.948 |
| OT | ΠGDM | 100 | 7.99 | 0.122 | 34.57 | 0.867 | 9.60 | 0.107 | 35.11 | 0.903 |
| VP-SDE | OT-ODE | 70 | **6.39** | **0.120** | **35.04** | **0.882** | 3.25 | **0.062** | **37.41** | **0.965** |
| VP-SDE | VP-ODE | 70 | 8.47 | 0.129 | 34.43 | 0.876 | 5.83 | 0.087 | 35.85 | 0.938 |
| VP-SDE | ΠGDM | 100 | 9.75 | 0.130 | 34.45 | 0.858 | 10.69 | 0.124 | 34.72 | 0.882 |
| VP-SDE | RED-Diff | 1000 | 14.63 | 0.171 | 32.42 | 0.820 | 9.19 | 0.105 | 33.52 | 0.895 |

Table 16: Quantitative evaluation of linear inverse problems on AFHQ-256 × 256

| Model | Inference | NFEs ↓ | SR 4×, $\sigma_y = 0.05$ | | | | Gaussian deblur, $\sigma_y = 0.05$ | | | |
|---|---|---|---|---|---|---|---|---|---|---|
| | | | FID ↓ | LPIPS ↓ | PSNR ↑ | SSIM ↑ | FID ↓ | LPIPS ↓ | PSNR ↑ | SSIM ↑ |
| OT | OT-ODE | 100 | **6.03** | **0.219** | **31.12** | **0.739** | **7.57** | **0.268** | **30.27** | **0.626** |
| OT | VP-ODE | 100 | 6.81 | 0.229 | 31.01 | 0.728 | 7.80 | 0.276 | 30.21 | 0.616 |
| OT | ΠGDM | 100 | 12.69 | 0.285 | 30.18 | 0.665 | 24.60 | 0.383 | 28.93 | 0.429 |
| VP-SDE | OT-ODE | 100 | **7.28** | **0.238** | 30.83 | 0.714 | **8.53** | **0.276** | **30.37** | 0.641 |
| VP-SDE | VP-ODE | 100 | 8.02 | 0.243 | **30.96** | **0.727** | 10.21 | 0.289 | 30.21 | 0.621 |
| VP-SDE | ΠGDM | 100 | 77.49 | 0.469 | 29.34 | 0.469 | 116.42 | 0.535 | 28.49 | 0.313 |
| VP-SDE | RED-Diff | 1000 | 20.84 | 0.331 | 29.97 | 0.675 | 15.81 | 0.341 | 30.15 | **0.645** |

Table 17: Quantitative evaluation of linear inverse problems on AFHQ-256 × 256

| Model | Inference | NFEs ↓ | Inpainting-*Center*, $\sigma_y = 0.05$ | | | | Denoising, $\sigma_y = 0.05$ | | | |
|---|---|---|---|---|---|---|---|---|---|---|
| | | | FID ↓ | LPIPS ↓ | PSNR ↑ | SSIM ↑ | FID ↓ | LPIPS ↓ | PSNR ↑ | SSIM ↑ |
| OT | OT-ODE | 100 | 8.98 | **0.104** | **35.32** | **0.897** | 2.48 | 0.061 | 37.18 | **0.965** |
| OT | VP-ODE | 100 | **7.48** | 0.107 | 35.02 | 0.892 | 3.38 | 0.075 | **37.41** | 0.954 |
| OT | ΠGDM | 100 | 19.09 | 0.153 | 34.20 | 0.855 | 22.87 | 0.237 | 32.93 | 0.823 |
| VP-SDE | OT-ODE | 100 | 9.93 | **0.107** | **35.18** | **0.892** | 2.17 | **0.060** | **37.95** | 0.963 |
| VP-SDE | VP-ODE | 100 | **8.78** | **0.107** | 35.12 | 0.891 | 3.08 | 0.071 | 37.68 | 0.959 |
| VP-SDE | ΠGDM | 100 | 57.46 | 0.239 | 32.40 | 0.773 | 81.15 | 0.451 | 29.62 | 0.639 |
| VP-SDE | RED-Diff | 1000 | 11.02 | 0.124 | 34.97 | 0.893 | 4.93 | 0.112 | 34.18 | 0.899 |

Table 18: Quantitative evaluation of linear inverse problems on face-blurred ImageNet-64 × 64

| Model | Inference | NFEs ↓ | SR 2×, $\sigma_y = 0$ | | | | Gaussian deblur, $\sigma_y = 0$ | | | |
|---|---|---|---|---|---|---|---|---|---|---|
| | | | FID ↓ | LPIPS ↓ | PSNR ↑ | SSIM ↑ | FID ↓ | LPIPS ↓ | PSNR ↑ | SSIM ↑ |
| OT | OT-ODE | 80 | **6.46** | 0.119 | 31.59 | 0.839 | **2.59** | **0.038** | 35.31 | 0.961 |
| OT | VP-ODE | 80 | 8.29 | 0.147 | 31.20 | 0.817 | 6.13 | 0.083 | 33.31 | 0.929 |
| OT | ΠGDM | 100 | 6.89 | **0.115** | **32.02** | **0.853** | 4.53 | 0.051 | **35.88** | **0.963** |
| VP-SDE | OT-ODE | 80 | **6.32** | 0.118 | 31.60 | 0.839 | **2.61** | **0.037** | 35.45 | 0.963 |
| VP-SDE | VP-ODE | 80 | 7.76 | 0.145 | 31.21 | 0.817 | 5.68 | 0.080 | 33.37 | 0.931 |
| VP-SDE | ΠGDM | 100 | 6.47 | **0.113** | **32.03** | **0.853** | 4.35 | 0.049 | **35.95** | **0.964** |
| VP-SDE | RED-Diff | 1000 | 11.74 | 0.224 | 30.12 | 0.798 | 15.39 | 0.134 | 31.99 | 0.879 |

Table 19: Quantitative evaluation of linear inverse problems on face-blurred ImageNet-$64 \times 64$

| Model | Inference | NFEs ↓ | Inpainting-*Center*, $\sigma_y = 0$ | | | |
|---|---|---|---|---|---|---|
| | | | FID ↓ | LPIPS ↓ | PSNR ↑ | SSIM ↑ |
| OT | OT-ODE | 80 | **4.94** | **0.080** | 37.42 | 0.885 |
| OT | VP-ODE | 80 | 7.85 | 0.120 | 34.24 | 0.858 |
| OT | ΠGDM | 100 | 6.09 | 0.082 | 36.75 | **0.901** |
| VP-SDE | OT-ODE | 80 | **4.85** | **0.079** | 37.64 | 0.887 |
| VP-SDE | VP-ODE | 80 | 7.21 | 0.117 | 34.33 | 0.860 |
| VP-SDE | ΠGDM | 100 | 5.79 | 0.081 | 36.81 | 0.902 |
| VP-SDE | RED-Diff | 1000 | 7.29 | **0.079** | **39.14** | **0.925** |

Table 20: Quantitative evaluation of linear inverse problems on face-blurred ImageNet-$128 \times 128$

| Model | Inference | NFEs ↓ | Inpainting-*Center*, $\sigma_y = 0$ | | | |
|---|---|---|---|---|---|---|
| | | | FID ↓ | LPIPS ↓ | PSNR ↑ | SSIM ↑ |
| OT | OT-ODE | 70 | 5.88 | **0.095** | **37.06** | 0.894 |
| OT | VP-ODE | 70 | 8.63 | 0.144 | 34.48 | 0.864 |
| OT | ΠGDM | 100 | **5.82** | 0.097 | 36.89 | **0.908** |
| VP-SDE | OT-ODE | 70 | 5.93 | 0.094 | 37.31 | 0.898 |
| VP-SDE | VP-ODE | 70 | 8.08 | 0.142 | 34.55 | 0.865 |
| VP-SDE | ΠGDM | 100 | 5.74 | 0.095 | 37.01 | 0.911 |
| VP-SDE | RED-Diff | 1000 | **5.40** | **0.068** | **38.91** | **0.928** |

Table 21: Quantitative evaluation of linear inverse problems on face-blurred ImageNet-$128 \times 128$

| Model | Inference | NFEs ↓ | SR 2×, $\sigma_y = 0$ | | | | Gaussian deblur, $\sigma_y = 0$ | | | |
|---|---|---|---|---|---|---|---|---|---|---|
| | | | FID ↓ | LPIPS ↓ | PSNR ↑ | SSIM ↑ | FID ↓ | LPIPS ↓ | PSNR ↑ | SSIM ↑ |
| OT | OT-ODE | 70 | **4.46** | **0.097** | 33.88 | 0.903 | **2.09** | **0.048** | 37.49 | **0.961** |
| OT | VP-ODE | 70 | 7.69 | 0.144 | 32.93 | 0.871 | 6.02 | 0.108 | 34.73 | 0.925 |
| OT | ΠGDM | 100 | 6.09 | 0.105 | **34.28** | **0.910** | 4.28 | 0.066 | **37.56** | **0.961** |
| VP-SDE | OT-ODE | 70 | 4.62 | 0.096 | 33.95 | 0.906 | **2.26** | **0.046** | **37.79** | **0.967** |
| VP-SDE | VP-ODE | 70 | 7.91 | 0.144 | 32.87 | 0.869 | 5.64 | 0.105 | 34.81 | 0.928 |
| VP-SDE | ΠGDM | 100 | 6.02 | 0.104 | 34.33 | 0.911 | 4.35 | 0.065 | 37.70 | 0.963 |
| VP-SDE | RED-Diff | 1000 | **3.90** | **0.082** | **34.47** | **0.92** | 4.19 | 0.085 | 34.68 | 0.929 |

Table 22: Quantitative evaluation of linear inverse problems on AFHQ-$256 \times 256$

| Model | Inference | NFEs ↓ | SR 4×, $\sigma_y = 0$ | | | | Gaussian deblur, $\sigma_y = 0$ | | | |
|---|---|---|---|---|---|---|---|---|---|---|
| | | | FID ↓ | LPIPS ↓ | PSNR ↑ | SSIM ↑ | FID ↓ | LPIPS ↓ | PSNR ↑ | SSIM ↑ |
| OT | OT-ODE | 100 | **5.75** | **0.169** | 32.25 | 0.792 | **6.63** | **0.213** | 31.29 | 0.722 |
| OT | VP-ODE | 100 | 6.14 | 0.194 | 31.93 | 0.773 | 7.38 | 0.231 | 31.10 | 0.705 |
| OT | ΠGDM | 100 | 8.89 | 0.173 | **32.57** | **0.812** | 9.78 | 0.209 | **31.54** | **0.743** |
| VP-SDE | OT-ODE | 100 | **6.58** | **0.178** | 32.18 | 0.789 | **8.24** | **0.226** | 31.21 | 0.717 |
| VP-SDE | VP-ODE | 100 | 8.00 | 0.225 | 31.48 | 0.742 | 9.19 | 0.252 | 30.91 | 0.688 |
| VP-SDE | ΠGDM | 100 | 10.85 | 0.189 | **32.52** | **0.811** | 11.46 | 0.228 | **31.47** | **0.738** |
| VP-SDE | RED-Diff | 1000 | 8.65 | 0.191 | 32.21 | 0.801 | 11.67 | 0.268 | 31.30 | 0.731 |

Table 23: Quantitative evaluation of linear inverse problems on AFHQ-$256 \times 256$

| Model | Inference | NFEs ↓ | Inpainting-*Center*, $\sigma_y = 0$ | | | | Inpainting-*Free-form*, $\sigma_y = 0$ | | | |
|---|---|---|---|---|---|---|---|---|---|---|
| | | | FID ↓ | LPIPS ↓ | PSNR ↑ | SSIM ↑ | FID ↓ | LPIPS ↓ | PSNR ↑ | SSIM ↑ |
| OT | OT-ODE | 100 | 8.87 | **0.061** | **37.45** | 0.921 | **4.98** | **0.097** | 36.15 | 0.889 |
| OT | VP-ODE | 100 | 9.18 | 0.106 | 35.63 | 0.898 | 6.92 | 0.135 | 34.72 | 0.869 |
| OT | ΠGDM | 100 | **7.36** | 0.080 | **37.45** | **0.933** | 6.52 | 0.100 | **36.58** | **0.913** |
| VP-SDE | OT-ODE | 100 | 9.95 | 0.064 | 37.49 | 0.918 | **5.39** | 0.099 | 36.15 | 0.887 |
| VP-SDE | VP-ODE | 100 | 10.50 | 0.112 | 35.59 | 0.893 | 7.36 | 0.139 | 34.65 | 0.865 |
| VP-SDE | ΠGDM | 100 | 8.61 | 0.088 | 37.27 | 0.925 | 7.25 | 0.109 | 36.37 | **0.906** |
| VP-SDE | RED-Diff | 1000 | **8.53** | **0.050** | **38.89** | **0.951** | 7.27 | **0.090** | **36.88** | 0.892 |

## D.2 ADDITIONAL QUALITATIVE RESULTS

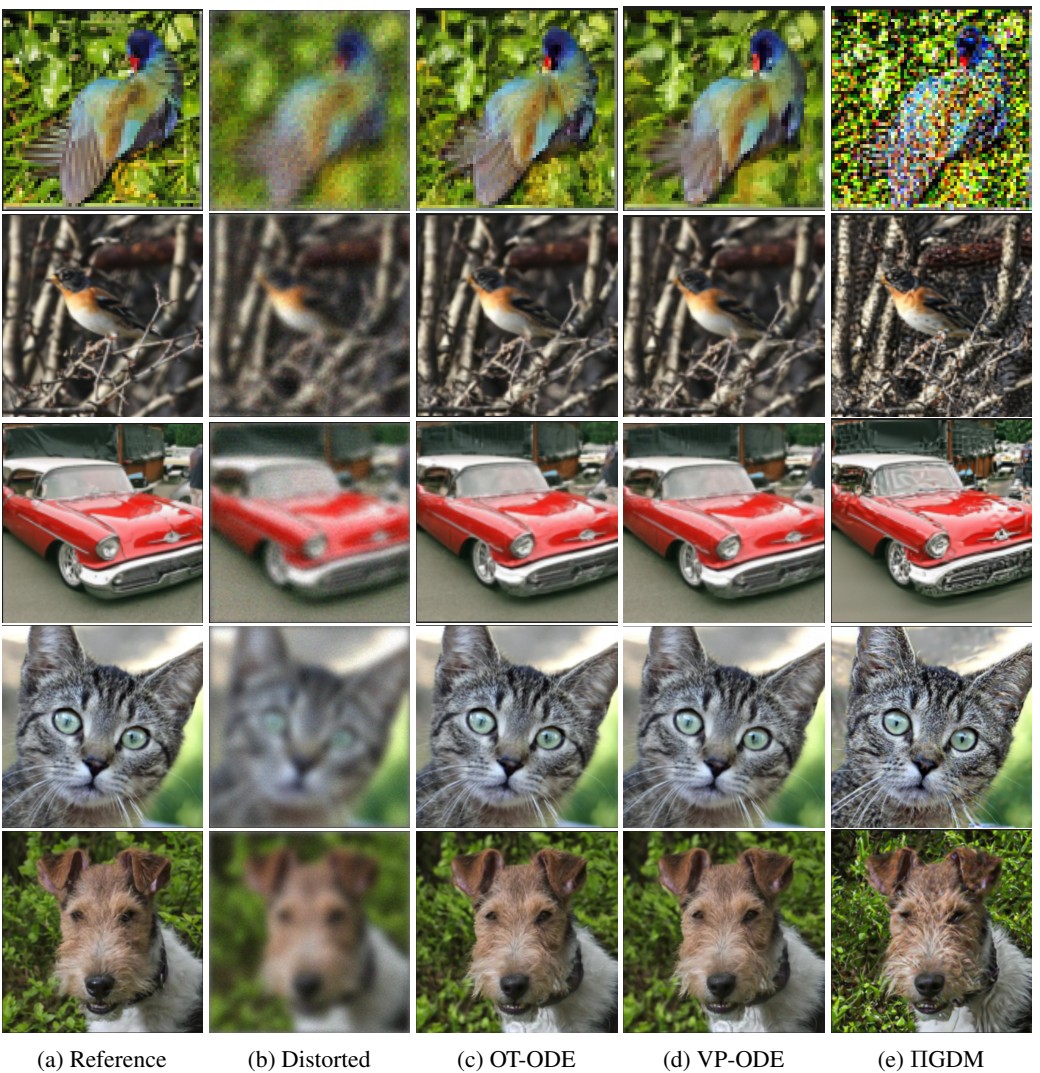

   (a) Reference       (b) Distorted       (c) OT-ODE       (d) VP-ODE       (e) ΠGDM

Figure 15: Gaussian-deblur with conditional OT model and $\sigma_y = 0.05$ for (**first row**) face-blurred ImageNet-64, (**second and third row**) face-blurred ImageNet-128, and ( **fourth and fifth row**) AFHQ.

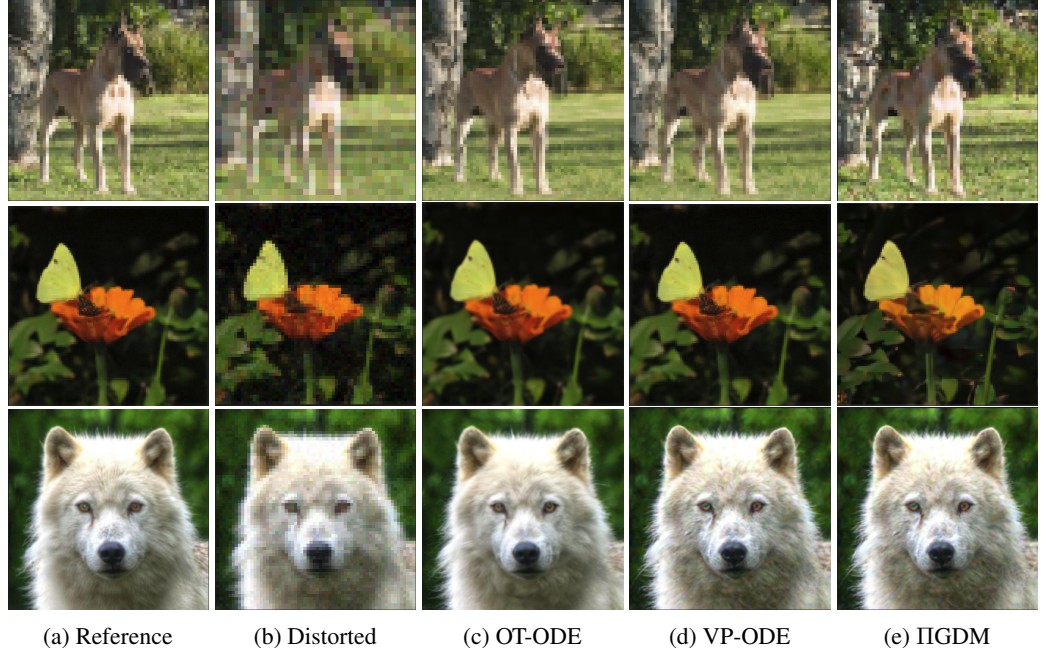

|  (a) Reference  |  (b) Distorted  |  (c) OT-ODE  |  (d) VP-ODE  |  (e) ΠGDM  |

Figure 16: Super-resolution with conditional OT model and $\sigma_y = 0.05$ for (**first row**) face-blurred ImageNet-64 $2\times$, (**second row**) face-blurred ImageNet-128 $2\times$, and (**third row**) AFHQ $4\times$.

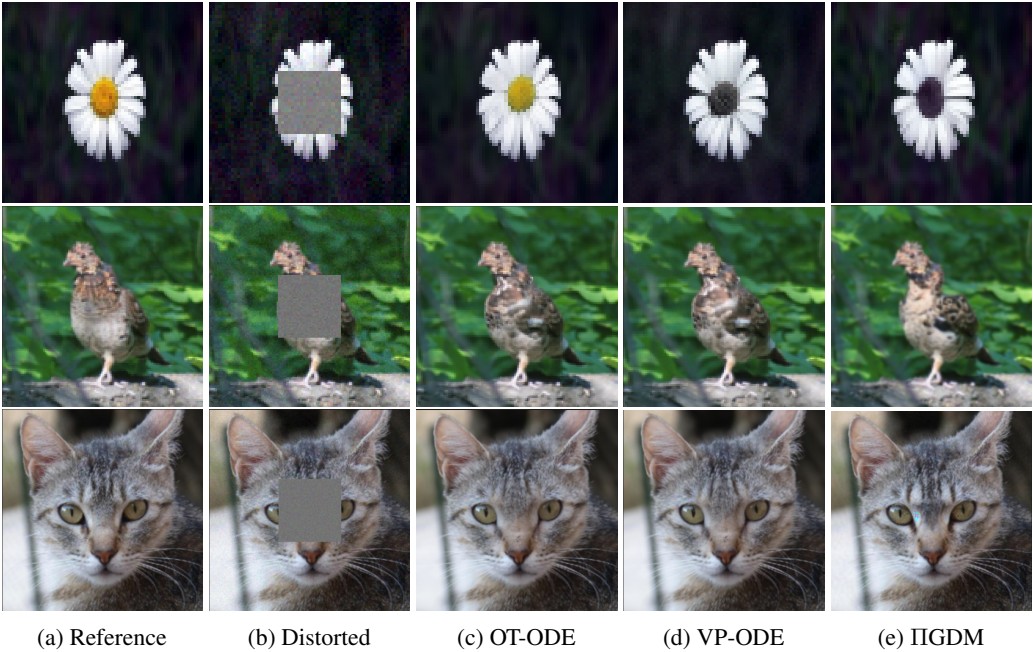

|  (a) Reference  |  (b) Distorted  |  (c) OT-ODE  |  (d) VP-ODE  |  (e) ΠGDM  |

Figure 17: Inpainting (Center mask) with conditional OT model and $\sigma_y = 0.05$ for (**first row**) face-blurred ImageNet-64, (**second row**) face-blurred ImageNet-128, and (**third row**) AFHQ.

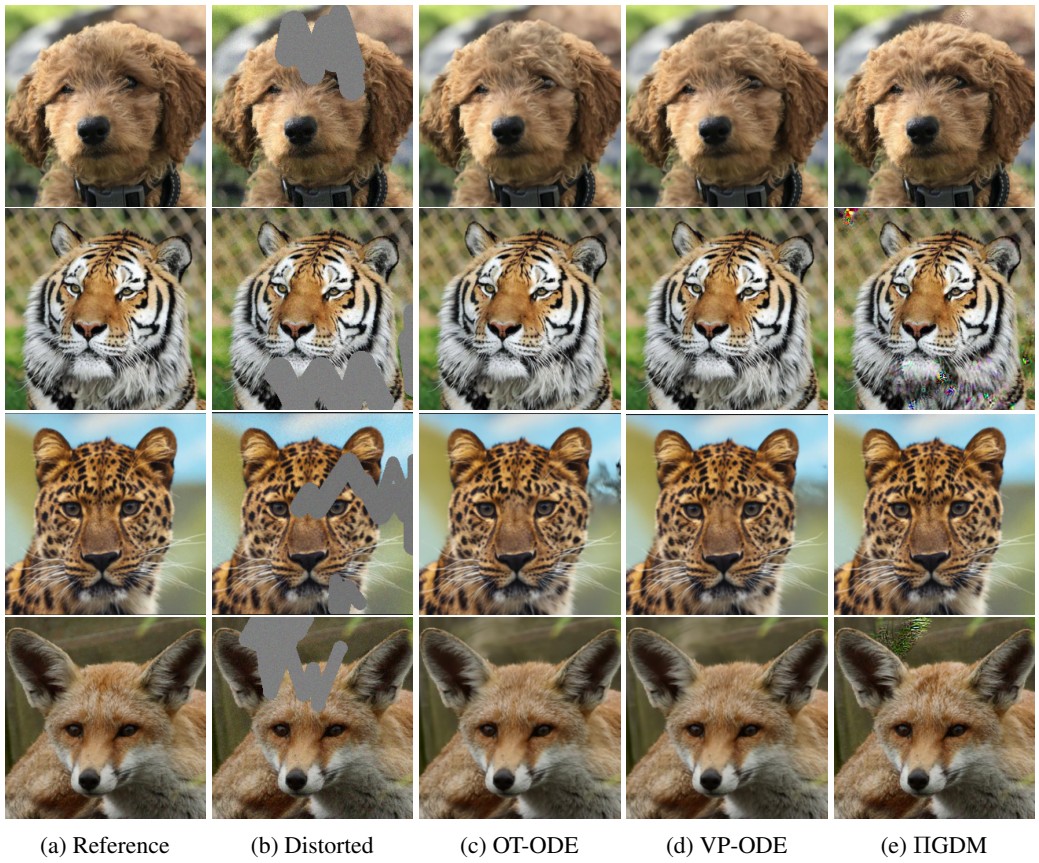

|      (a) Reference      |      (b) Distorted      |      (c) OT-ODE      |      (d) VP-ODE      |      (e) ΠGDM      |

Figure 18: Inpainting (Free-form mask) with conditional OT model and $\sigma_y = 0.05$ for AFHQ.

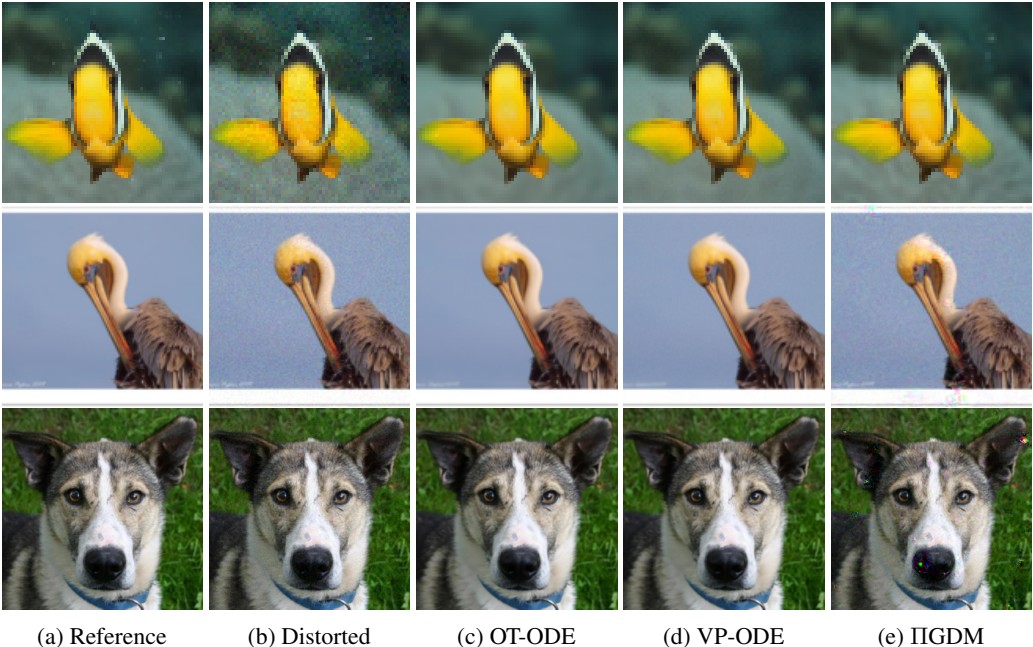

|      (a) Reference      |      (b) Distorted      |      (c) OT-ODE      |      (d) VP-ODE      |      (e) ΠGDM      |

Figure 19: Denoising with conditional OT model and $\sigma_y = 0.05$ for (**first row**) face-blurred ImageNet-64, (**second row**) face-blurred ImageNet-128, and (**third row**) AFHQ.

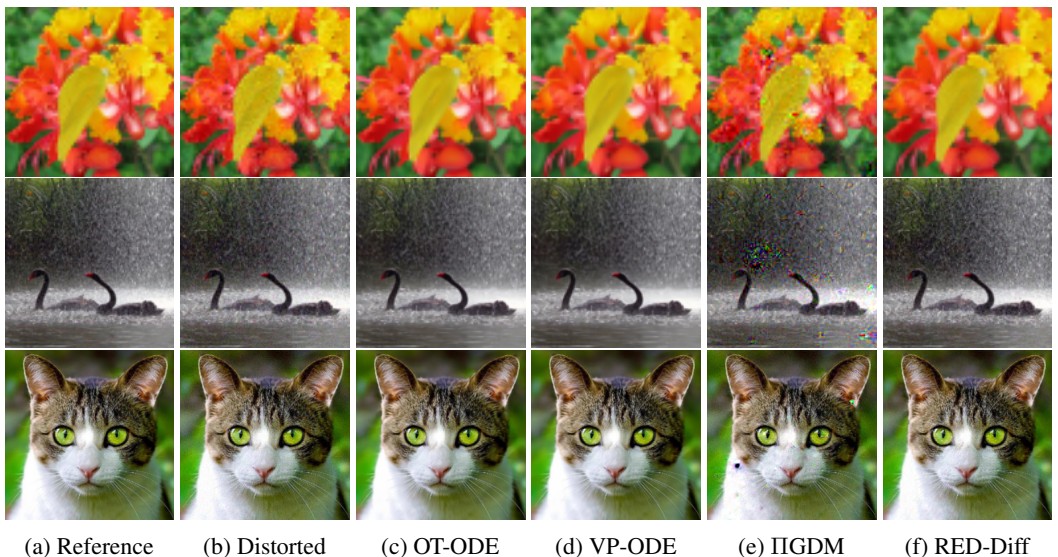

| (a) Reference | (b) Distorted | (c) OT-ODE | (d) VP-ODE | (e) ΠGDM | (f) RED-Diff |

Figure 20: Denoising with pretrained VP-SDE model and $\sigma_y = 0.05$ for (**first row**) face-blurred ImageNet-64, (**second row**) face-blurred ImageNet-128, and (**third row**) AFHQ.

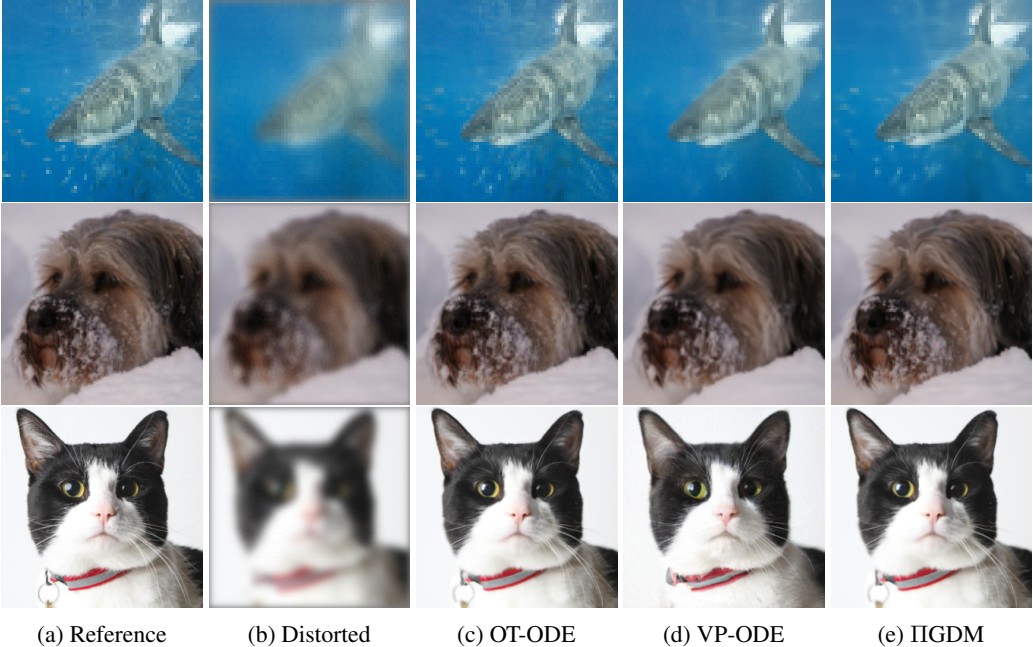

| (a) Reference | (b) Distorted | (c) OT-ODE | (d) VP-ODE | (e) ΠGDM |

Figure 21: Gaussian deblurring with conditional OT model and $\sigma_y = 0$ for (**first row**) face-blurred ImageNet-64, (**second row**) face-blurred ImageNet-128 and (**third row**) AFHQ.

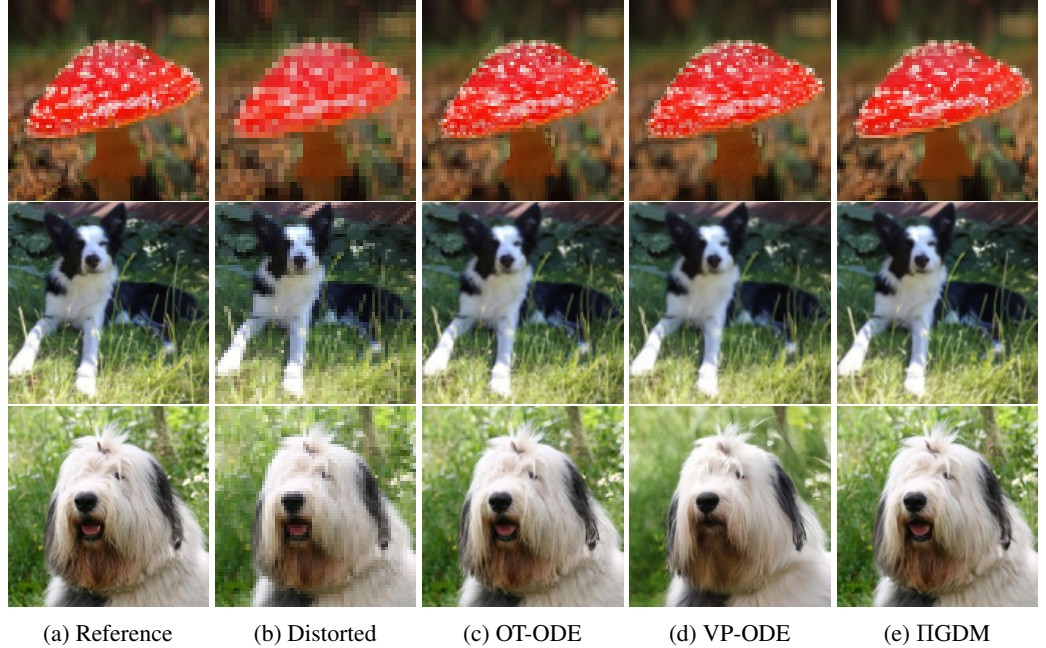

|               |               |            |            |           |
|---------------|---------------|------------|------------|-----------|
| (a) Reference | (b) Distorted | (c) OT-ODE | (d) VP-ODE | (e) ΠGDM  |

Figure 22: Super-resolution with conditional OT model and $\sigma_y = 0$ for (**first row**) face-blurred ImageNet-64, (**second row**) face-blurred ImageNet-128 and (**third row**) AFHQ.

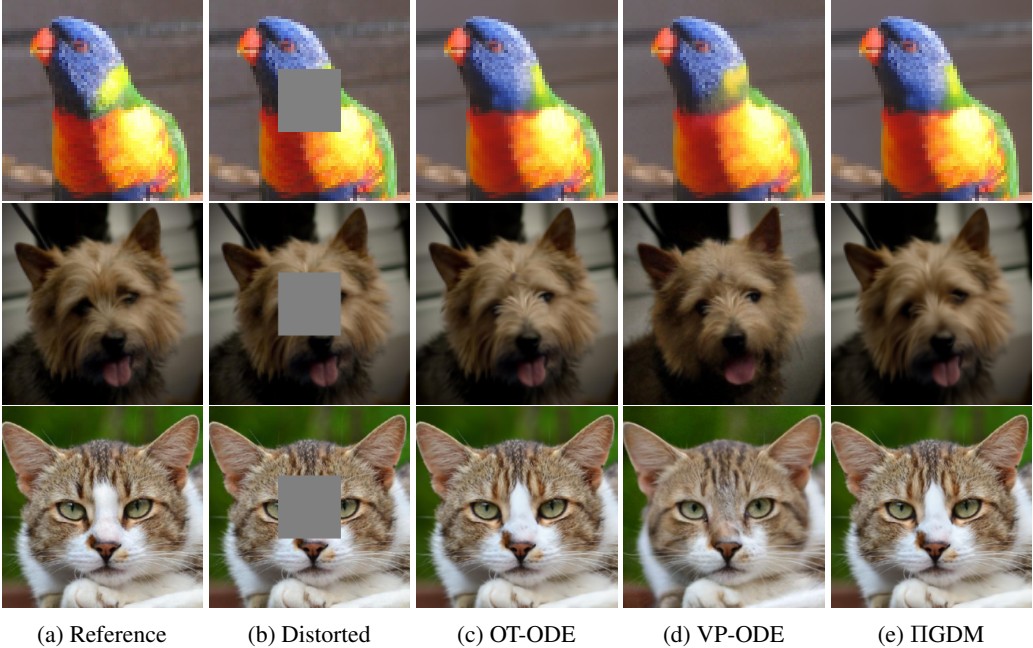

|               |               |            |            |           |
|---------------|---------------|------------|------------|-----------|
| (a) Reference | (b) Distorted | (c) OT-ODE | (d) VP-ODE | (e) ΠGDM  |

Figure 23: Inpainting (centered mask) with conditional OT model and $\sigma_y = 0$ for (**first row**) face-blurred ImageNet-64, (**second row**) face-blurred ImageNet-128 and (**third row**) AFHQ.

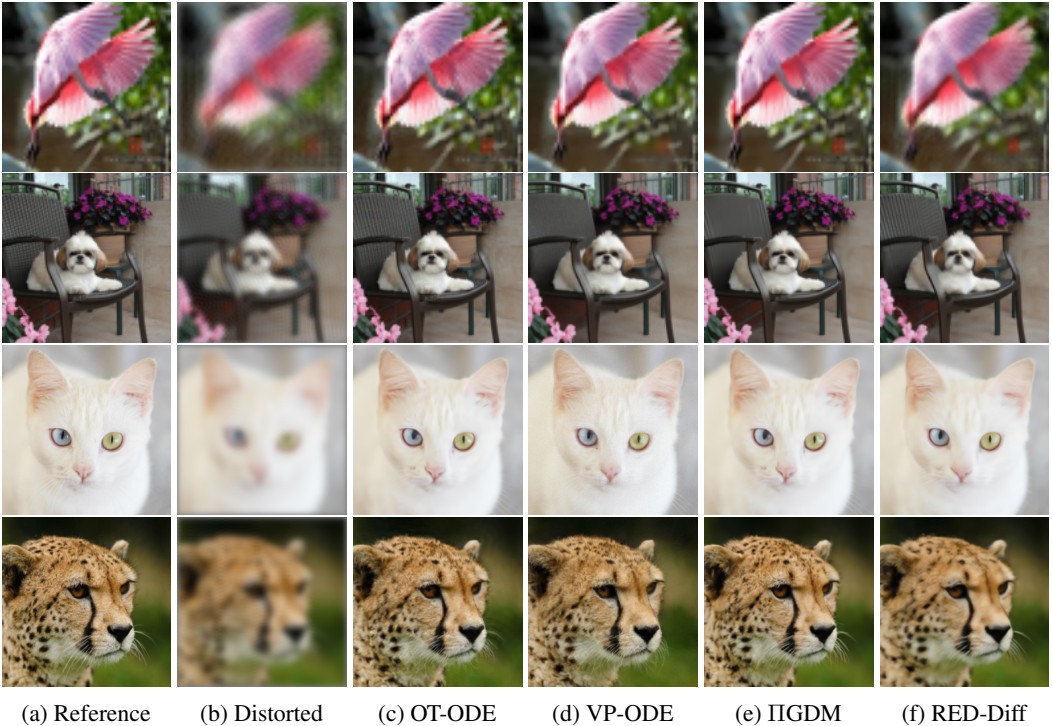

|  (a) Reference | (b) Distorted | (c) OT-ODE | (d) VP-ODE | (e) ΠGDM | (f) RED-Diff |

Figure 24: Gaussian deblurring with VP-SDE model and $\sigma_y = 0$ for (**first row**) face-blurred ImageNet-64, (**second row**) face-blurred ImageNet-128 and (**third and fourth row**) AFHQ.

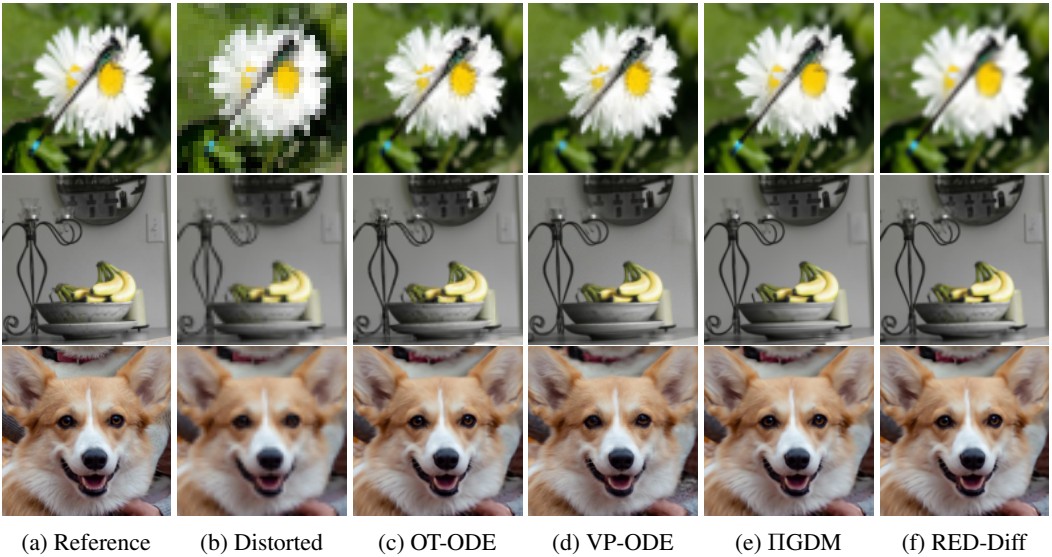

|  (a) Reference | (b) Distorted | (c) OT-ODE | (d) VP-ODE | (e) ΠGDM | (f) RED-Diff |

Figure 25: Super-resolution with VP-SDE model and $\sigma_y = 0$ for (**first row**) face-blurred ImageNet-64, (**second row**) face-blurred ImageNet-128 and (**third row**) AFHQ.

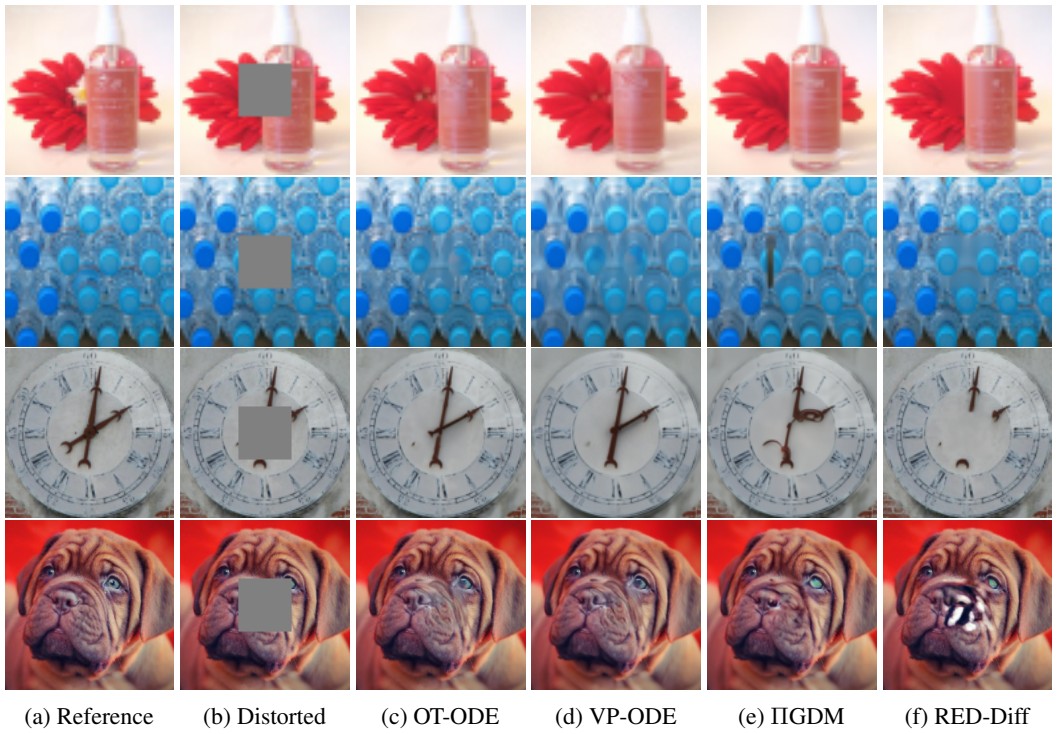

|  (a) Reference | (b) Distorted | (c) OT-ODE | (d) VP-ODE | (e) ΠGDM | (f) RED-Diff |

Figure 26: Inpainting (centered mask) with VP-SDE model and $\sigma_y = 0$ for (**first and second row**) face-blurred ImageNet-64, (**third row**) face-blurred ImageNet-128 and (**fourth row**) AFHQ.

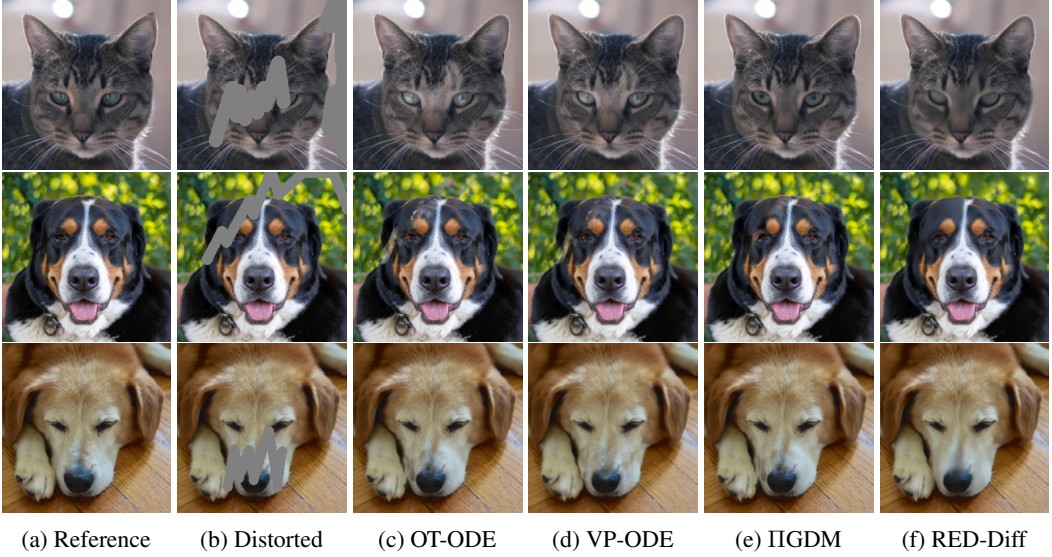

|  (a) Reference | (b) Distorted | (c) OT-ODE | (d) VP-ODE | (e) ΠGDM | (f) RED-Diff |

Figure 27: Inpainting (freeform mask) with VP-SDE model and $\sigma_y = 0$ for AFHQ.

# E  BASELINES

## E.1  ΠGDM

**Implementation details.**    We closely follow the official code available on github while implementing ΠGDM. For noisy case, we closely follow the Algorithm 1 in the appendix of Song et al. (2022). We use adaptive weighted guidance for both noiseless and noisy cases as in the original work. We always use uniform spacing while iterating the timestep over 100 steps. We use ascending time from 0 to 1. Note that the original paper uses descending time from $T$ to 0. According to the notational convention used in this paper, this is equivalent to ascending time from 0 to 1. For the choice of $r_t^2$, we consider the values derived from both variance exploding formulation and variance preserving formulation.

**Value of $r_t^2$.**    ΠGDM sets the value of $r_t^2 = \frac{\sigma_{1-t}^2}{1+\sigma_{1-t}^2}$ for VE-SDE, where $q(\boldsymbol{x}_t|\boldsymbol{x}_1) = \mathcal{N}(\boldsymbol{x}_1, \sigma_{1-t}^2 \boldsymbol{I})$. We can follow the same procedure as outlined in Song et al. (2022), and solve for $r_t^2$ in closed form for VP-SDE. We know for that VP-SDE, $q(\boldsymbol{x}_t|\boldsymbol{x}_1) = \mathcal{N}(\alpha_{1-t}\boldsymbol{x}_1, (1-\alpha_{1-t}^2)\boldsymbol{I})$, where $\alpha_t = e^{-\frac{1}{2}T(t)}$, $T(t) = \int_0^t \beta(s)ds$, and $\beta(s)$ is the noise scale function. Using equation 13 for VP-SDE gives $r_t^2 = 1 - \alpha_{1-t}^2$. We can also obtain an alternate $r_t^2$ by plugging in value of $\sigma_t^2$ for VP-SDE into the expression of $r_t^2$ derived for VE-SDE, which evaluates to $r_t^2 = \frac{1-\alpha_{1-t}^2}{2-\alpha_{1-t}^2}$. Empirically, we find that $r_t^2$ for VE-SDE marginally outperforms VP-SDE. We report performance of ΠGDM with both choices of $r_t^2$ in Table  24 to 26.

Table 24: Relative performance of ΠGDM on face-blurred ImageNet-64 with VE and VP derived $r_t^2$ with $\sigma_y = 0.05$

| Measurement | Model | VP | | | | VE | | | |
|---|---|---|---|---|---|---|---|---|---|
| | | FID ↓ | LPIPS ↓ | PSNR ↑ | SSIM ↑ | FID ↓ | LPIPS ↓ | PSNR ↑ | SSIM ↑ |
| SR 2× | OT | 6.52 | 0.168 | 30.54 | 0.753 | 5.91 | 0.160 | 30.60 | 0.762 |
| Gaussian deblur | OT | 55.19 | 0.374 | 28.74 | 0.516 | 39.36 | 0.326 | 29.00 | 0.572 |
| Inpainting-*Center* | OT | 9.25 | 0.111 | 34.13 | 0.863 | 8.70 | 0.109 | 34.17 | 0.864 |
| Denoising | OT | 16.59 | 0.102 | 34.60 | 0.906 | 16.44 | 0.101 | 34.64 | 0.907 |
| SR 2× | VP-SDE | 6.84 | 0.174 | 30.48 | 0.743 | 6.11 | 0.166 | 30.54 | 0.753 |
| Gaussian deblur | VP-SDE | 54.77 | 0.376 | 28.74 | 0.511 | 39.14 | 0.329 | 28.99 | 0.567 |
| Inpainting-*Center* | VP-SDE | 9.75 | 0.113 | 34.03 | 0.860 | 9.36 | 0.112 | 34.06 | 0.862 |
| Denoising | VP-SDE | 17.19 | 0.107 | 34.25 | 0.901 | 15.54 | 0.102 | 34.41 | 0.906 |

Table 25: Relative performance of ΠGDM on face-blurred ImageNet-128 with VE and VP derived $r_t^2$ with $\sigma_y = 0.05$

| Measurement | Model | VP | | | | VE | | | |
|---|---|---|---|---|---|---|---|---|---|
| | | FID ↓ | LPIPS ↓ | PSNR ↑ | SSIM ↑ | FID ↓ | LPIPS ↓ | PSNR ↑ | SSIM ↑ |
| SR 2× | OT | 4.38 | 0.148 | 32.07 | 0.831 | 4.26 | 0.145 | 32.12 | 0.834 |
| Gaussian deblur | OT | 30.30 | 0.328 | 29.96 | 0.606 | 22.42 | 0.296 | 30.17 | 0.642 |
| Inpainting-*Center* | OT | 7.99 | 0.122 | 34.57 | 0.867 | 7.64 | 0.120 | 34.61 | 0.869 |
| Denoising | OT | 9.60 | 0.107 | 35.11 | 0.903 | 9.30 | 0.104 | 35.21 | 0.906 |
| SR 2× | VP-SDE | 7.55 | 0.183 | 31.61 | 0.785 | 6.14 | 0.168 | 31.79 | 0.803 |
| Gaussian deblur | VP-SDE | 55.61 | 0.463 | 28.57 | 0.414 | 41.69 | 0.404 | 28.98 | 0.493 |
| Inpainting-*Center* | VP-SDE | 9.75 | 0.130 | 34.45 | 0.858 | 9.46 | 0.129 | 34.49 | 0.859 |
| Denoising | VP-SDE | 10.69 | 0.124 | 34.72 | 0.882 | 10.11 | 0.119 | 34.92 | 0.886 |

**Choice of starting time.**    For OT-ODE sampling and VP-ODE sampling, we observe that starting at time $t > 0$ improves the performance. We therefore perform an ablation study on ΠGDM baseline, and vary the start time to verify whether starting at $t > 0$ helps to improve the performance. We plot the metrics for three different measurements in  Figure 28. We observe that starting later at time $t > 0$ consistently leads to worse performance compared to starting at time $t = 0$. Therefore, for all our experiments with ΠGDM, we always start at time $t = 0$.

Table 26: Relative performance of $\Pi$GDM on AFHQ with VE and VP derived $r_t^2$ with $\sigma_y = 0.05$

| Measurement | Model | VP | | | | VE | | | |
|---|---|---|---|---|---|---|---|---|---|
| | | FID ↓ | LPIPS ↓ | PSNR ↑ | SSIM ↑ | FID ↓ | LPIPS ↓ | PSNR ↑ | SSIM ↑ |
| SR 4× | OT | 12.69 | 0.285 | 30.18 | 0.665 | 12.31 | 0.282 | 30.23 | 0.672 |
| Gaussian deblur | OT | 24.60 | 0.383 | 28.93 | 0.429 | 19.66 | 0.355 | 29.16 | 0.475 |
| Inpainting-*Center* | OT | 19.09 | 0.153 | 34.20 | 0.855 | 16.51 | 0.145 | 34.40 | 0.863 |
| Denoising | OT | 11.20 | 0.159 | 34.49 | 0.876 | 10.92 | 0.153 | 34.78 | 0.883 |
| SR 4× | VP-SDE | 77.49 | 0.469 | 29.34 | 0.469 | 54.12 | 0.413 | 29.73 | 0.549 |
| Gaussian deblur | VP-SDE | 116.42 | 0.535 | 28.49 | 0.313 | 95.09 | 0.493 | 28.74 | 0.368 |
| Inpainting-*Center* | VP-SDE | 57.46 | 0.239 | 32.40 | 0.773 | 56.86 | 0.238 | 32.42 | 0.775 |
| Denoising | VP-SDE | 81.15 | 0.451 | 29.62 | 0.639 | 35.33 | 0.278 | 31.72 | 0.776 |

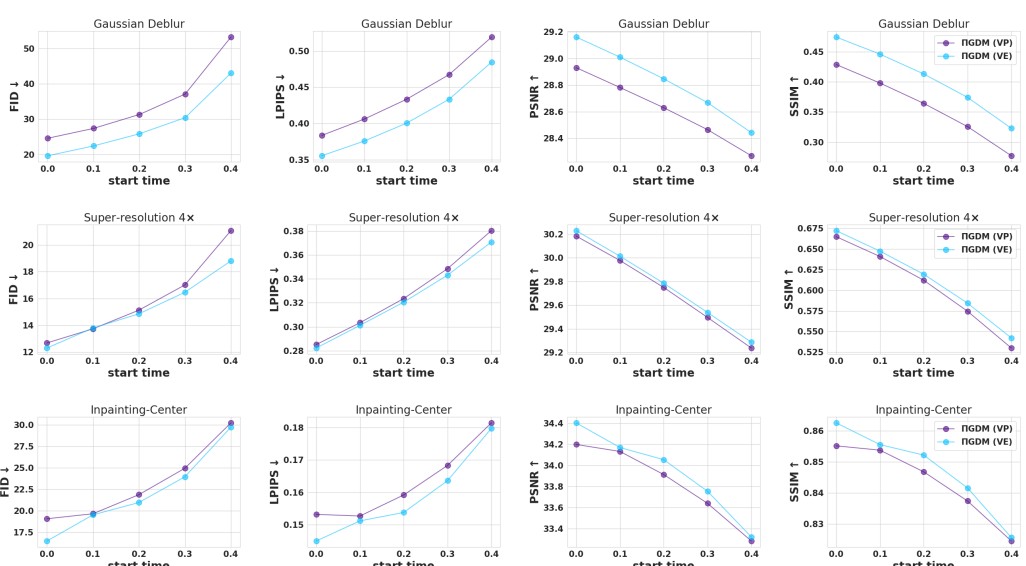

Figure 28: Variation in performance $\Pi$GDM sampling with variation in start times on AFHQ dataset. We use pretrained conditional OT model and set $\sigma_y = 0.05$. We observe similar trends with VP-SDE checkpoint. We plot metrics for both choices of $r_t^2$ that can be derived from variance preserving and variance exploding formulations.

## E.2 RED-DIFF

**Implementation details.** We use VP-SDE model for all experiments with RED-Diff. We closely follow the official code available on github while implementing RED-Diff. Similar to (Mardani et al., 2023), we always use uniform spacing while iterating the timestep over 1000 steps. We use ascending time from 0 to 1. Note that the original paper uses descending time from $T$ to 0. According to the notational convention used in this paper, this is equivalent to ascending time from 0 to 1. We use Adam optimizer and use the momentum pair $(0.9, 0.99)$ similar to the original work. Further, we use initial learning rate of 0.1 for AFHQ and ImageNet-128, as used in the original work, and learning rate of 0.01 for ImageNet-64. We use batch size of 1 for all the experiments. Finally, we extensively tuned the regularization hyperparameter $\lambda$ to find the value that results in optimal performance across all metrics. We summarize the results of our experiments in Table 27 to 32. We note that more extensive tuning may be able to find better performing hyperparameters but this goes against the intent of a training-free algorithm.

Table 27: Hyperparameter search for RED-Diff on face-blurred ImageNet-$64 \times 64$ with $\sigma_y = 0.05$. We use learning rate of $0.01$.

| $\lambda$ | SR 2×, $\sigma_y = 0.05$ | | | | Gaussian deblur, $\sigma_y = 0.05$ | | | |
|---|---|---|---|---|---|---|---|---|
| | FID ↓ | LPIPS ↓ | PSNR ↑ | SSIM ↑ | FID ↓ | LPIPS ↓ | PSNR ↑ | SSIM ↑ |
| 0.1 | 34.09 | 0.224 | 30.12 | 0.798 | **46.76** | 0.254 | 29.29 | 0.715 |
| 0.25 | 28.45 | 0.206 | 30.40 | 0.814 | 51.20 | **0.236** | **30.19** | **0.776** |
| 0.75 | **23.02** | **0.187** | **31.22** | **0.839** | 73.76 | 0.287 | 30.47 | 0.750 |
| 1.5 | 32.35 | 0.243 | 30.80 | 0.792 | 82.26 | 0.335 | 30.29 | 0.705 |
| 2.0 | 40.33 | 0.284 | 30.41 | 0.750 | 86.48 | 0.358 | 30.17 | 0.683 |
| $\lambda$ | Inpainting-*Center*, $\sigma_y = 0.05$ | | | | Denoising, $\sigma_y = 0.05$ | | | |
| | FID ↓ | LPIPS ↓ | PSNR ↑ | SSIM ↑ | FID ↓ | LPIPS ↓ | PSNR ↑ | SSIM ↑ |
| 0.1 | 15.71 | 0.155 | 31.74 | 0.840 | 12.47 | 0.085 | 32.24 | 0.907 |
| 0.25 | 15.56 | 0.155 | 31.73 | 0.839 | 11.80 | 0.083 | 32.36 | 0.908 |
| 0.75 | 13.31 | 0.139 | 32.65 | 0.857 | 8.43 | 0.062 | 33.65 | 0.932 |
| 1.5 | **12.18** | **0.119** | 33.97 | 0.881 | 6.11 | 0.041 | 35.34 | 0.958 |
| 2.0 | 12.87 | **0.119** | **34.19** | **0.886** | **6.02** | **0.041** | **35.64** | **0.964** |

Table 28: Hyperparameter search for RED-Diff on face-blurred ImageNet-$64 \times 64$ with $\sigma_y = 0$. We use learning rate of $0.01$.

| $\lambda$ | SR 2×, $\sigma_y = 0$ | | | | Gaussian deblur, $\sigma_y = 0$ | | | |
|---|---|---|---|---|---|---|---|---|
| | FID ↓ | LPIPS ↓ | PSNR ↑ | SSIM ↑ | FID ↓ | LPIPS ↓ | PSNR ↑ | SSIM ↑ |
| 0.1 | **11.74** | 0.224 | 30.12 | 0.798 | **15.39** | **0.134** | **31.99** | **0.879** |
| 0.25 | 12.65 | **0.130** | **32.34** | **0.886** | 29.56 | 0.236 | 30.19 | 0.776 |
| 0.75 | 20.36 | 0.187 | 31.22 | 0.839 | 55.43 | 0.287 | 30.47 | 0.750 |
| 1.5 | 33.13 | 0.243 | 30.80 | 0.792 | 71.64 | 0.335 | 30.29 | 0.705 |
| 2.0 | 41.56 | 0.288 | 30.46 | 0.752 | 78.55 | 0.358 | 30.22 | 0.685 |

| $\lambda$ | Inpainting-*Center*, $\sigma_y = 0$ | | | |
|---|---|---|---|---|
| | FID ↓ | LPIPS ↓ | PSNR ↑ | SSIM ↑ |
| 0.1 | **7.29** | **0.079** | **39.14** | **0.925** |
| 0.25 | 7.40 | 0.155 | 31.73 | 0.839 |
| 0.75 | 8.47 | 0.083 | 38.59 | 0.922 |
| 1.5 | 10.75 | 0.095 | 37.42 | 0.916 |
| 2.0 | 12.54 | 0.119 | 34.19 | 0.886 |

Table 29: Hyperparameter search for RED-Diff on face-blurred ImageNet-$128 \times 128$ with $\sigma_y = 0.05$. We use learning rate of $0.1$.

| $\lambda$ | SR 2×, $\sigma_y = 0.05$ | | | | Gaussian deblur, $\sigma_y = 0.05$ | | | |
|---|---|---|---|---|---|---|---|---|
| | FID ↓ | LPIPS ↓ | PSNR ↑ | SSIM ↑ | FID ↓ | LPIPS ↓ | PSNR ↑ | SSIM ↑ |
| 0.1 | 23.25 | 0.272 | 30.12 | 0.731 | 37.83 | 0.42 | 28.54 | 0.473 |
| 0.75 | 14.56 | 0.224 | 30.71 | 0.782 | **21.43** | **0.229** | 31.41 | 0.807 |
| 1.5 | **10.54** | **0.182** | 31.82 | 0.852 | 22.85 | 0.247 | **31.65** | **0.809** |
| 2.0 | 11.65 | 0.187 | **31.93** | **0.859** | 24.71 | 0.259 | 31.61 | 0.802 |
| $\lambda$ | Inpainting-*Center*, $\sigma_y = 0.05$ | | | | Denoising, $\sigma_y = 0.05$ | | | |
| | FID ↓ | LPIPS ↓ | PSNR ↑ | SSIM ↑ | FID ↓ | LPIPS ↓ | PSNR ↑ | SSIM ↑ |
| 0.1 | 19.68 | 0.191 | 31.75 | 0.795 | 12.83 | 0.134 | 32.27 | 0.854 |
| 0.75 | 19.03 | 0.202 | 31.36 | 0.779 | 12.69 | 0.14 | 32.09 | 0.846 |
| 1.5 | 16.33 | 0.189 | 31.81 | 0.794 | 10.67 | 0.121 | 32.89 | 0.874 |
| 2.0 | **14.63** | **0.171** | **32.42** | **0.819** | **9.19** | **0.105** | **33.52** | **0.895** |

Table 30: Hyperparameter search for RED-Diff on face-blurred ImageNet-$128 \times 128$ with $\sigma_y = 0$. We use learning rate of $0.1$.

| $\lambda$ | SR $2\times$, $\sigma_y = 0$ | | | | Gaussian deblur, $\sigma_y = 0$ | | | |
|---|---|---|---|---|---|---|---|---|
| | FID $\downarrow$ | LPIPS $\downarrow$ | PSNR $\uparrow$ | SSIM $\uparrow$ | FID $\downarrow$ | LPIPS $\downarrow$ | PSNR $\uparrow$ | SSIM $\uparrow$ |
| 0.1 | **3.90** | **0.082** | **34.47** | **0.922** | **4.19** | **0.085** | **34.68** | **0.929** |
| 0.75 | 6.52 | 0.105 | 33.54 | 0.905 | 12.59 | 0.177 | 32.71 | 0.864 |
| 1.5 | 10.46 | 0.142 | 32.98 | 0.894 | 19.29 | 0.225 | 32.15 | 0.831 |
| 2.0 | 13.08 | 0.165 | 32.65 | 0.884 | 22.57 | 0.245 | 31.94 | 0.816 |
| $\lambda$ | Inpainting-*Center*, $\sigma_y = 0$ | | | | Inpainting-*Freeform*, $\sigma_y = 0$ | | | |
| | FID $\downarrow$ | LPIPS $\downarrow$ | PSNR $\uparrow$ | SSIM $\uparrow$ | FID $\downarrow$ | LPIPS $\downarrow$ | PSNR $\uparrow$ | SSIM $\uparrow$ |
| 0.1 | **5.39** | **0.068** | **38.91** | **0.928** | **8.94** | **0.162** | **35.54** | **0.830** |
| 0.75 | 5.52 | 0.073 | 38.11 | 0.924 | 9.26 | 0.166 | 35.05 | 0.826 |
| 1.5 | 6.09 | 0.079 | 37.32 | 0.920 | 10.13 | 0.172 | 34.58 | 0.821 |
| 2.0 | 6.68 | 0.083 | 36.87 | 0.917 | 10.87 | 0.176 | 34.30 | 0.818 |

Table 31: Hyperparameter search for RED-Diff on AFHQ with $\sigma_y = 0.5$. We use learning rate of $0.1$.

| $\lambda$ | SR $4\times$, $\sigma_y = 0.05$ | | | | Gaussian deblur, $\sigma_y = 0.05$ | | | |
|---|---|---|---|---|---|---|---|---|
| | FID $\downarrow$ | LPIPS $\downarrow$ | PSNR $\uparrow$ | SSIM $\uparrow$ | FID $\downarrow$ | LPIPS $\downarrow$ | PSNR $\uparrow$ | SSIM $\uparrow$ |
| 0.1 | 21.59 | 0.385 | 29.51 | 0.607 | 17.36 | 0.379 | 29.95 | 0.639 |
| 0.25 | 22.47 | 0.374 | 29.66 | 0.635 | **15.81** | **0.341** | **30.15** | **0.645** |
| 0.75 | **20.84** | **0.331** | **29.97** | **0.675** | 25.41 | 0.366 | 29.76 | 0.588 |
| 1.5 | 22.46 | 0.355 | 29.68 | 0.642 | 38.66 | 0.409 | 29.34 | 0.525 |
| 2.0 | 25.02 | 0.376 | 29.49 | 0.618 | 45.01 | 0.427 | 29.18 | 0.500 |
| $\lambda$ | Inpainting-*Center*, $\sigma_y = 0.05$ | | | | Denoising, $\sigma_y = 0.05$ | | | |
| | FID $\downarrow$ | LPIPS $\downarrow$ | PSNR $\uparrow$ | SSIM $\uparrow$ | FID $\downarrow$ | LPIPS $\downarrow$ | PSNR $\uparrow$ | SSIM $\uparrow$ |
| 0.1 | **28.39** | 0.216 | 31.53 | 0.756 | 8.32 | 0.159 | 32.18 | 0.827 |
| 0.25 | 28.85 | 0.217 | 31.51 | 0.755 | 8.35 | 0.161 | 32.16 | 0.826 |
| 0.75 | 28.80 | 0.218 | 31.64 | 0.759 | 7.94 | 0.156 | 32.35 | 0.833 |
| 1.5 | 28.74 | 0.205 | 32.19 | 0.784 | 6.63 | 0.138 | 33.12 | 0.862 |
| 2.0 | 28.55 | 0.190 | 32.63 | 0.802 | 5.71 | 0.124 | 33.70 | 0.882 |
| 2.5 | 28.71 | **0.177** | **32.99** | **0.818** | **4.93** | **0.111** | **34.18** | **0.899** |

Table 32: Hyperparameter search for RED-Diff on AFHQ with $\sigma_y = 0$. We use learning rate (lr) of $0.1$ unless mentioned otherwise.

| $\lambda$ | SR $4\times$, $\sigma_y = 0$ | | | | Gaussian deblur, $\sigma_y = 0$ | | | |
|---|---|---|---|---|---|---|---|---|
| | FID $\downarrow$ | LPIPS $\downarrow$ | PSNR $\uparrow$ | SSIM $\uparrow$ | FID $\downarrow$ | LPIPS $\downarrow$ | PSNR $\uparrow$ | SSIM $\uparrow$ |
| 0.005 | 11.67 | 0.197 | **32.93** | **0.837** | 14.69 | 0.278 | **31.73** | **0.760** |
| 0.05 | **8.65** | **0.191** | 32.21 | 0.801 | 11.67 | **0.268** | 31.30 | 0.731 |
| 0.1 | 9.65 | 0.204 | 31.84 | 0.781 | **11.53** | 0.273 | 31.05 | 0.711 |
| 0.25 | 11.65 | 0.222 | 31.53 | 0.768 | 13.22 | 0.293 | 30.63 | 0.675 |
| 0.75 | 14.98 | 0.274 | 30.72 | 0.726 | 23.34 | 0.351 | 29.91 | 0.598 |
| 1.5 | 19.40 | 0.332 | 29.95 | 0.665 | 36.96 | 0.402 | 29.39 | 0.529 |
| 2.0 | 22.72 | 0.361 | 29.65 | 0.632 | 43.64 | 0.422 | 29.22 | 0.504 |
| $\lambda$ | Inpainting-*Center*, $\sigma_y = 0$, lr=0.01 | | | | Inpainting-*Freeform*, $\sigma_y = 0$ | | | |
| | FID $\downarrow$ | LPIPS $\downarrow$ | PSNR $\uparrow$ | SSIM $\uparrow$ | FID $\downarrow$ | LPIPS $\downarrow$ | PSNR $\uparrow$ | SSIM $\uparrow$ |
| 0.005 | **8.53** | **0.050** | **38.89** | **0.951** | 7.22 | 0.091 | **36.89** | **0.892** |
| 0.05 | 8.53 | 0.050 | 38.89 | 0.951 | 7.27 | **0.090** | 36.88 | **0.892** |
| 0.1 | 8.53 | 0.050 | 38.88 | 0.951 | **7.23** | 0.091 | 36.82 | 0.891 |
| 0.25 | 8.53 | 0.050 | 38.83 | 0.950 | 7.32 | 0.094 | 36.69 | 0.889 |
| 0.75 | 8.88 | 0.056 | 38.60 | 0.948 | 7.74 | 0.102 | 36.26 | 0.884 |
| 1.5 | 10.32 | 0.071 | 38.04 | 0.942 | 8.41 | 0.112 | 35.69 | 0.877 |
| 2.0 | 11.62 | 0.084 | 37.54 | 0.937 | 8.76 | 0.119 | 35.37 | 0.872 |

# F  ADDITIONAL BACKGROUND

In this section, we follow the notation used in the prior work by Lipman et al. (2022).

**Continuous Normalizing Flows (CNFs).**  A Continuous Normalizing Flow (Chen et al., 2018a) is a time-dependent diffeomorphic map $\phi_t : [0,1] \times \mathbb{R}^d \to \mathbb{R}^d$ that is defined by the ODE:

$$\frac{d}{dt}\phi_t(\boldsymbol{x}) = \boldsymbol{v}_t(\phi_t(\boldsymbol{x})); \quad \phi_0(\boldsymbol{x}) = \boldsymbol{x} \tag{19}$$

where $\boldsymbol{x} \in \mathbb{R}^d$ and $\boldsymbol{v}_t : [0,1] \times \mathbb{R}^d \to \mathbb{R}^d$ is a time-dependent vector field that is usually parametrized with a neural network. The generative process of a CNF involves sampling from a simple prior distribution $\boldsymbol{x}_0 \sim p_0(\boldsymbol{x}_0)$ (e.g. standard Gaussian distribution) and then solving the initial value problem defined by the ODE in Eq. (19) to obtain a sample from the target distribution $\boldsymbol{x}_1 \sim p_1(\boldsymbol{x}_1)$. Thus, a CNF reshapes a simple prior distribution $p_0$ to a more complex distribution $p_t$, via a push-forward equation based on the instantaneous change of variables formula.

$$p_t = [\phi]_* p_0 \tag{20}$$

$$[\phi]_* p_0(\boldsymbol{x}) = p_0(\phi_t^{-1}(\boldsymbol{x})) \det\left[\frac{\partial \phi_t^{-1}}{\partial \boldsymbol{x}}(\boldsymbol{x})\right] \tag{21}$$

CNFs are usually trained by optimizing the maximum likelihood objective. As shown in Chen et al. (2018a), the exact likelihood computation can be done via relatively cheap operations despite the Jacobian term. However, this requires restricting the architecture of the neural network to constrain the Jacobian term. FFJORD (Grathwohl et al., 2018) improves upon this by proposing a method that uses Hutchinson's trace estimator to compute log density, and allows CNFs with free-form Jacobians, thereby removing any restrictions on the architecture. This approach has difficulties for high-dimensional images where the trace estimator is noisy. Flow Matching provides an alternative, scalable approach to training CNFs with arbitrary architectures.

**Flow Matching.**  Suppose we have samples from an unknown data distribution $\boldsymbol{x}_1 \sim q(\boldsymbol{x}_1)$. Let $p_t$ denote a probability path from the prior distribution $p_0$ to the data distribution $p_1$ that is approximately equal to $q$. Flow Matching loss is defined as

$$\mathcal{L}_{FM} = \mathbb{E}_{t,p_t(\boldsymbol{x})} \|\boldsymbol{v}_t(\boldsymbol{x}; \theta) - \boldsymbol{u}_t(\boldsymbol{x})\|^2 \tag{22}$$

where $\boldsymbol{u}_t(\boldsymbol{x})$ is a vector field that generates the probability path $p_t(\boldsymbol{x})$, and $\theta$ denotes trainable parameters of the CNF. In practice, we usually do not have any prior knowledge on $p_t$ and $u_t$, and thus this objective is intractable. Inspired by diffusion models, Lipman et al. (2022) propose Conditional Flow Matching, where both the probability paths and the vector fields are conditioned on the sample $\boldsymbol{x}_1 \sim q(\boldsymbol{x}_1)$. The exact objective for Conditional Flow matching is given by

$$\mathcal{L}_{CFM} = \mathbb{E}_{t,q(\boldsymbol{x}_1),p_t(\boldsymbol{x}|\boldsymbol{x}_1)} \|\boldsymbol{v}_t(\boldsymbol{x}; \theta) - \boldsymbol{u}_t(\boldsymbol{x}|\boldsymbol{x}_1)\|^2 \tag{23}$$

where, $p_t(\boldsymbol{x}|\boldsymbol{x}_1)$ denotes a conditional probability path, and $\boldsymbol{u}_t(\boldsymbol{x}|\boldsymbol{x}_1)$ denotes the corresponding conditional vector field that generates the conditional probability path. Interestingly, both the loss objectives in Eq. (23) and Eq. (22) have identical gradients w.r.t. $\theta$. More importantly, past research has proven that $\boldsymbol{u}_t(\boldsymbol{x}) = \mathbb{E}[\boldsymbol{u}_t(\boldsymbol{x}|\boldsymbol{x}_1)|\boldsymbol{x}_t = \boldsymbol{x}]$. The optimal solution to the conditional Flow Matching recovers $\boldsymbol{u}_t(\boldsymbol{x})$ and therefore $\boldsymbol{v}_t(\boldsymbol{x}; \theta)$ generates the desired probability path $p_t(\boldsymbol{x})$. Thus, we can train a CNF without access to the marginal vector field $\boldsymbol{u}_t(\boldsymbol{x})$ or probability path $p_t(\boldsymbol{x})$. Compared to the prior approaches to train flow models, Flow Matching allows simulation-free training with unbiased gradients, and scales easily to high dimensions.

