# OpenReview forum: "Training-free Linear Image Inversion via Flows"
_ICLR.cc/2024/Conference — ICLR 2024 Conference Withdrawn Submission_

### Official Review · Reviewer_pako · 2023-10-15

**Soundness:** 3 good
**Presentation:** 3 good
**Contribution:** 2 fair
**Rating:** 6
**Confidence:** 5

**Summary:**

The paper proposes a flow matching model (or equivalently, diffusion model-based) based inverse problem solver. Unlike previous diffusion model-based approaches which are based on stochastic samplers, the proposed method is based on ODE integration. $\Pi$GDM approximation is used to estimate the time-dependent log likelihood is used. The method is shown to outperform previous arts ($\Pi$GDM, RED-Diff) on some datasets: ImageNet-64/128, and AFHQ-256.

**Strengths:**

1. To the best of my knowledge, this work is the first to use flow matching models (although they can be thought of as equivalent to diffusion models) to solve linear inverse problems.

2. The method is easy to understand, given that it builds on the prior approximation of $\Pi$GDM.

**Weaknesses:**

1. (Limited contribution) As stated in S1, the proposed method boils down to $\Pi$GDM that uses ODE sampler, rather than the stochastic DDIM sampler ($\eta = 1.0$) that $\Pi$GDM uses. The step size derived from Lemma 2 is also another way of stating what was already shown from DDRM and $\Pi$GDM.

2. (Choice of test dataset) On most of the diffusion model-based inverse problem solvers, testing is performed on two canonical datasets: FFHQ 256$\times$256 and ImageNet 256$\times$256. These two datasets are typically more challenging than the datasets that are used in this paper. It is hard to convince the strength of the paper when there is no specific reason to deviate from this standard. The method would be more convincing if the same superiority can be seen in such largely used benchmarks.

3. (Exposition) In the preliminaries, it is unclear why one has to start from *conditional* diffusion models and *conditional* flow models, when at the end of the day, unconditional models will be used to try to sample from the posterior distribution $p(\mathbf{x}_1|\mathbf{y})$.

**Questions:**

1. The authors repeatedly use the term *image inversion* throughout the work. Instead, I would advise to use the term *inverse problems in imaging*, which is a standard term that has been used for decades.

2. Is Algorithm 1 different from $\Pi$GDM other than the fact that it is an ODE sampler? Specifically, assume that $\eta = 0.0$ from the $\Pi$GDM DDIM sampling. Are these two equivalent in this case, given that VP-ODE is used?

3. (Implementation details) It is unclear why the authors trained a class conditional model, where the standard is to use class unconditional models. The former is known to perform better than the latter.

4. Why does VP-ODE perform (marginally) better than OT-ODE? The theory of flow matching models would state otherwise.

5. (pg 9. last paragraph) Did you mean to cite DPS when you mention non-linear observations? PSLD would not be able to solve non-linear inverse problems due to the gluing term used in the paper.

---

> ### Author Response · Authors · 2023-11-15
>
> Thank you for your thoughtful feedback on our work. We have tried our best to address your questions and concerns below.  We are certain that our method is sufficiently distinct and hope our response clarifies its novelty and significance sufficient for acceptance.
>
> 1. We first emphasize that our method for solving linear inverse problems, as presented in Algorithm 1, is __not__ $\Pi$GDM that uses a VP-ODE (DDIM) sampler.  We expand upon and enumerate the differences below. Instead, Algorithm 1 outlines our method for solving inverse problems via conditional Optimal Transport (OT) paths with a pretrained diffusion model. As demonstrated by our experiments, images restored via the conditional OT probability path outperform both VP-ODE paths and $\Pi$GDM. We also emphasize Lemma 2 is _not equivalent_ to the previous results on scaling intermediate noisy images $x_t$ in DDRM and $\Pi$GDM.
>
> **Key differences between our method and $\Pi$GDM**
>
> - **Different sampling paths for solving inverse problems**:   Our algorithm employs conditional OT probability paths as opposed to VE-SDE/VP-SDE probability paths employed by $\Pi$GDM for solving linear inverse problems. VE-SDE and VP-SDE models are characterized by specific schedules for $\alpha_t$ and $\sigma_t$, and these parameters are coupled through the use of a $\beta_t$ schedule. In contrast, the OT schedule uses functions for $\alpha_t$ and $ \sigma_t$ that cannot be modeled by either VE-SDE or VP-SDE. This key distinction in our approach of using conditional OT probability path enables our method to outperform diffusion paths across a wide range of linear inverse problems and datasets, as demonstrated in our paper.
>
> - **Early starting**: Our algorithm starts sampling from time $t > 0$ while $\Pi$GDM starts at $t=0$ (Gaussian noise).  As can be seen in the supplementary material in Figure 7 and Figure 8, early starting is beneficial to our method, but not beneficial to $\Pi$GDM as can be seen in Figure 28.
>
> - **Reduced hyperparameter tuning**: The use of conditional OT path significantly reduces the amount of manual hyperparameter tuning. The only hyperparameter in our case is the choice of start time, that we keep constant for all tasks and datasets.
>     - Our method exhibits more stable performance across a wide range of NFEs compared to both VP-ODE and  $\Pi$GDM. Further, our method outperforms both VP-ODE and $\Pi$GDM at a smaller number of function evaluations (NFEs) as shown in Figure 6.
>
>     - In contrast to $\Pi$GDM, our method does not incorporate any guided weights for solving linear inverse problems via conditional OT paths. We find that the guidance weights used in $\Pi$GDM hurt performance on conditional OT paths.
>
> - Our algorithm focuses on correcting the vector fields of flow models, as opposed to the  correction of intermediate noisy images typically done by diffusion models.
>
> - Experiments with our VP-ODE method, an alternative to our OT-ODE also introduced in our paper, could be considered similar to that of the $\Pi$GDM algorithm with $\eta=0.0$.  We point out though that $\Pi$GDM explored different values of $\eta$ in their Appendix (see their Table 6 and 7) and found that $\eta=1.0$ was superior at 100 NFE’s, which is the number of NFE’s used by $\Pi$GDM in our paper.  This is completely different from our results where our VP-ODE method performed better than $\Pi$GDM with the recommended $\eta=1.0$ for solving noisy linear inverse problems.  So while there are similarities, our VP-ODE method and $\Pi$GDM at $\eta=0.0$ are different and not the same as suggested by the referee at $\eta=0.0$.  Further, we emphasize that our recommended OT-ODE method is even more conceptually different and performs even better than our VP-ODE method.
>
> Overall, our algorithm for solving linear inverse problems works for flow models with arbitrary affine Gaussian probability paths (Algorithm 1,2,and 3) while $\Pi$GDM is specifically designed for VP-SDE/VE-SDE paths.

---

> ### Author Response · Authors · 2023-11-15
>
> **Key contributions**
>
> We identify that there is a gap in the literature for solving inverse problems via flow models. Flow models trained via flow matching naturally consider a  broader range of probability paths than diffusion models, and therefore, leveraging them  opens  the possibility to use optimal transport paths for solving inverse problems.
>
> - Our primary methodological contribution is Theorem 1, which provides the mathematical expression to correct the vector field $\hat{v}(x)$ of unconditional flow models with affine Gaussian probability paths. This allows us to retrieve the conditional vector field $\hat{v}(x,y)$ that can sample from the correct conditional posterior distribution $q(x_t|y)$.
>
> - We utilize Theorem 1 to propose an algorithm (See Algorithm 3) for solving linear inverse problems via flows with arbitrary affine Gaussian probability paths. We utilize $\Pi$GDM correction to estimate the intractable term $\nabla_{x_t} \log q(y | x_t)$ that arises in our expression to correct the unconditional vector field.
> - We empirically demonstrate that conditional OT paths outperform diffusion probability paths in solving linear inverse problems on a broad range of datasets (ImageNet 64/128 and AFHQ 256) and tasks (Gaussian deblurring, super-resolution, denoising and inpainting).
> - As pretrained flow models are not widely available, we offer a way to convert between flow models and diffusion models, including crucially between arbitrary Gaussian probability paths (Lemma 2).  This allows applying our OT-ODE method to diffusion models trained under any affine Gaussian probability path, as demonstrated by our results using our VP-SDE model checkpoint.  Algorithm 1 in the main paper outlines our method for solving linear inverse problems via conditional OT probability paths with a pretrained diffusion model. Additionally, Algorithm 2, found in the supplementary, enables use of pretrained diffusion models for solving linear inverse problems via flows with any affine Gaussian probability paths.
>
> In summary, our work offers a unifying algorithm for solving linear inverse problems, applicable to a broad range of flow models trained via flow matching as well as any existing pre-trained continuous-time diffusion models.
>
> **Distinction of Lemma 2 from prior works**:
> Lemma 2 provides a way to translate between arbitrary affine Gaussian probability paths $q$ and $q’$. This involves two key steps: 1) Evaluating the time where the Signal-to-Noise Ratio (SNR) of the two probability paths $q$ and $q’$ match. 2) Using the pretrained flow model/diffusion model to predict $\hat{x}_1$, where the model uses the converted time from step (1) and the noisy image is also rescaled (See Eq. 9 for the scaling factor.).
>
> This is very different from the limited method followed in $\Pi$GDM and DDRM, which converts VP-SDE ($p_t(x_t|x_0) =\mathcal{N}(\sqrt{\alpha_t}x_0, (1 - \alpha_t)I$) to VE-SDE ($p_t(x_t|x_0) = \mathcal{N}(x_0, \sigma_t^2I)$), where it is required that $\alpha_t = 1 / (1 + \sigma_t^2)$ (see Appendix B of the DDRM paper). In other words, with a given VP-SDE model, one can only utilize the past method to convert to VE-SDE with this particular noise schedule $\sigma_t$.  This requirement in fact insists that the SNR is equal at time $t$ under the two paths and hence leads to $t==t’$.
>
> Our approach on the other hand does not require this equal time limitation and works for any arbitrary pair of affine Gaussian probability paths including for example when $\sigma_t^2$ does not satisfy the requirement used by $\Pi$GDM and DDRM.  Depending on $\alpha_t$ and $\sigma_t$, our method results in different values of $t$ and $t’$ when converting VP-SDE to VE-SDE, and vice versa.  Further, conversion between the probability paths of OT-ODE and VP-ODE requires our formulation in Lemma 2 that utilizes different times.  Lemma 2 applies to any pair of affine Gaussian probability paths, not a particular pair of paths that satisfies $t==t’$ through an unnecessary requirement that SNRs match at all times.

---

> ### Author Response · Authors · 2023-11-15
>
> 2. **Choice of test datasets**: Due to ethical and privacy concerns associated with the use of real people’s faces in both FFHQ and ImageNet, we opted to use AFHQ, which has animal faces instead of FFHQ as a close relative, and face-blurred ImageNet. Because we needed to train models from scratch on these alternative datasets with better privacy, we used 128X128 images to save compute. We also include results on AFHQ 256X256. Overall, we include results on three different image resolutions (64X64, 128X128, and 256X256) and our results are consistent across datasets and tasks.
> 3. **Exposition**:  Our theoretical derivations require an understanding of what the optimal solution  should be for conditional diffusion and flow models, so that we can understand how to convert unconditional diffusion and flow models into meeting those expectations.  Our approximation is about _reducing the gap_ between the unconditional model and the conditional model that we would have liked to train.  It would not be possible to write about the optimal conditional vector field without this additional notation and background.
> 4. **Replace the phrase 'image inversion' with 'inverse problems for imaging' in the paper**: Thank you for this suggestion. We have updated our paper to use inverse problems for imaging instead of image inversion. As per ICLR policy, we cannot update the title of the paper during discussion phase, so we leave the original title unchanged.
>
> 5. **Implementation details**: We believe that the standard of using unconditional models does not correspond well to practice. Image generative models are often conditioned on text as well as other inputs. Our empirical results, by using class-conditional models, are therefore closer to practice which we believe increases their relevance. We did explore unconditional models at lower resolutions, and found similar trends in the results. Finally, all the methods in our paper used exactly the same model checkpoint, and therefore, there was no unfairness.
>
> 6. **Performance of VP-ODE against OT-ODE**: We find that our approach for solving linear  inverse problems with OT-ODE performs better than VP-ODE on both the model checkpoints. We note that it is not completely fair to compare across our two model checkpoints, as one of the models could have been trained for longer. We believe that for a fair comparison, we should only compare different methods for solving inverse problems on the same model checkpoint.
>
> 7. **Clarification about the page 9 last paragraph citation**: Here, we meant that linear image restoration problems become non-linear when applied to latent diffusion models. It is indeed true that the observations $y$ themselves are still linear in the pixel space, but the observations $y$ can be considered non-linear observations with respect to the latent space used in the latent diffusion. We have updated the paper to make this clarification.
>
> We are happy to answer any follow up questions.

---

> ### Comment · Area_Chair_qJsW · 2023-11-21
>
> Reviwer pako,
>
> Since the authors' had submitted the rebuttal with detailed answers to your concerns, and your original score was diverged from the others, could you please reply to the rebuttal and share your opinion?
>
> Thanks,
> AC

---

> > ### Comment · Reviewer_pako · 2023-11-22
> >
> > I would like to thank the authors for clarifying most of the concerns. I have updated my score accordingly. Apologies for the late update.

---

### Official Review · Reviewer_tnhX · 2023-10-31

**Soundness:** 3 good
**Presentation:** 3 good
**Contribution:** 3 good
**Rating:** 8
**Confidence:** 4

**Summary:**

They propose a training-free method for image inversion using pretrained flow models. Their approach leverages the simplicity and efficiency of Flow Matching models, significantly reducing the need for manual tuning. They draw inspiration from prior gradient correction methods and conditional Optimal Transport paths. Empirically, their flow-based method improves upon diffusion-based linear inversion methods across various noisy linear image inversion problems on high-dimensional datasets.

**Strengths:**

- Theoretical analysis is conducted based on linear corruption regarding conditional flow matching.
- Extensive experiments are performed to validate the proposed method from various perspectives.

**Weaknesses:**

- I wonder whether the linear inversion in Sec3 can handle a wide range of data corruption in the real world.

I believe this paper adequately addresses the problem and conducts various experiments. Overall, I think it is a good paper and I would like to accept it. Honest speaking, I am starting to use it as my new baseline for another submission.

**Questions:**

as above.

---

> ### Author Response · Authors · 2023-11-15
>
> Thank you for your extremely positive feedback on our work! We are excited to know that you are already using our method as a baseline!
>
> As the reviewer has rightly pointed out, it is indeed true that linear inverse problems cannot handle many data corruptions of the real world and we leave this extension as future work. We however note that our method can handle composition of linear corruptions. Further, we list potential directions to extend this work to non-linear observations in the conclusion. Finally, we emphasize that a strength of our method is that it is simple and requires essentially no hyperparameter tuning to get good results.

---

### Official Review · Reviewer_STfT · 2023-11-01

**Soundness:** 4 excellent
**Presentation:** 4 excellent
**Contribution:** 4 excellent
**Rating:** 8
**Confidence:** 3

**Summary:**

This paper proposes a training-free method for linear image inversion using pretrained flow models. The proposed method has theoretically-justified weighting schemes and thus require less hyperparameter-tuning. The authors show effectiveness of the proposed method on common high-dimensional datasets and compare to prior diffusion-based linear inversion methods.

**Strengths:**

1. The paper is clear and organized well. The visualized results are impressive.
2. The proposed method is well-motivated and is justified theoretically. Requiring less hyperparameter tuning is an important practical advantage for inverse problems.
3. Experiments is generally thorough and solid. Results reported on ImageNet and AFHQ demonstrated the effectiveness of the proposed method comparing to prior works.

**Weaknesses:**

[Minor]
1. In the paper, it seems DPS is assumed to have worse performance than ΠGDM as "ΠGDM is improved upon DPS". Empirically, is it also observed that DPS has worse performance than ΠGDM and the proposed method?
2. In Figure 2, the x-axis seems to be not continuous, but adjacent points are connected and interpolated with lines, which might be not intuitive.

**Questions:**

In Figure 5, ΠGDM shows green-dot artifacts, is this common for ΠGDM or rare cases?

---

> ### Author Response · Authors · 2023-11-15
>
> Thank you for your encouraging feedback on our work! We are happy to know that you find our paper well-motivated and theoretically justified, and that our experiments are thorough.  We have tried our best to answer your questions below:
>
> 1. DPS works well with a large number of NFEs (~1000 NFEs) but performs worse than $\Pi$GDM with few NFEs (Table 10, 11, 12 in the original $\Pi$GDM paper). As we are working in the regime of few NFEs, we use $\Pi$GDM which has stronger performance. Furthermore, $\Pi$GDM and DPS are closely related as their approximations differ by a scaling factor.
>
> 2. We agree that the styling choice for plotting figures might seem counterintuitive, and bar plots might be a more reasonable choice. We decided to plot it in this way because we felt the results are easier to interpret. The corresponding bar plot for Figure 2 would have $13 \times 4=52$ vertical bars in each plot which can be tricky to interpret. On the other hand, in the current plot, it is relatively easy to interpret that the red line, that corresponds to OT-ODE, outperforms other methods by always being above (or below) the competing methods.
>
> 3. We find that green-dot artifacts mostly occur in few images in $\Pi$GDM for noisy inverse problems but are relatively uncommon in the noiseless case. We presume that these dots occur because the model samples outside the support of images while solving the inverse problem.
>
> Again, thank you for your thoughtful questions and we are happy to answer any further questions on our work.

---

### Official Review · Reviewer_4dCg · 2023-11-01

**Soundness:** 3 good
**Presentation:** 2 fair
**Contribution:** 3 good
**Rating:** 8
**Confidence:** 3

**Summary:**

This paper explores the application of flow models to linear inverse problems. It specifically extends recently introduced techniques from diffusion models to flow models and offers a method for converting diffusion models into flow models. The paper also conducts a comprehensive set of experiments to validate these propositions.

**Strengths:**

While many papers have recently concentrated on approaches that utilize diffusion models for inverse problems, there appears to be a noticeable gap in the literature concerning the application of flow models in this context. This paper aims to bridge that gap. The extensive numerical results appear to suggest that their approach outperforms diffusion models, although it's worth noting that, at times, distinguishing differences or identifying major issues in the produced images can be challenging.

**Weaknesses:**

The paper's main weakness lies in its writing. It appears that Section 2, labeled 'Preliminaries,' and the rest of the paper are written with the assumption that readers are already well-versed in flow-based models trained with flow matching. This might create challenges for readers who are not familiar with this background. Considering the relative novelty of flow matching, particularly in comparison to diffusion methods, it would be beneficial for the authors to provide some background information before transitioning to the conditional case

**Questions:**

- Could the authors offer some insight into why having probability paths that are straighter than diffusion paths might be advantageous for reconstruction in inverse problems?

---

> ### Author Response · Authors · 2023-11-15
>
> Thank you for your encouraging comments on our work. We have tried our best to address your questions and concerns below:
>
> - _Additional background on flow matching_: We tried to include as much background information on flow matching as possible in the main paper but due to the constraints of space, we were unable to include extensive details. We will update  our paper to include additional background information on flow matching in the Supplementary material.
>
> - _Why straighter probability paths might outperform diffusion paths?_  If we trained a conditional flow matching model, we anticipate that straighter probability paths would lead to simpler training and better sampling. In the training-free case, we were motivated to consider conditional OT probability paths based on this intuition from the training perspective. Aligned with this, we observe straighter paths might potentially help with reducing the number of model evaluations needed for image restoration. As shown in Figure 6 in the Appendix, conditional OT paths have more stable performance than diffusion paths across a wide range of NFEs. Further, we achieve good performance in as few as 20 NFEs, and outperform both $\Pi$GDM and VP-ODE sampling.
>
> We are happy to answer any follow-up questions about our work.

---

> > ### Author Response · Authors · 2023-11-18
> >
> > Dear Reviewer 4dCg,
> >
> > We have updated our paper to include additional background information on flow matching. These details can be found in the Appendix F.
> >
> > Please reach out to us if you have any further questions.

---

> > > ### Comment · Reviewer_4dCg · 2023-11-19
> > >
> > > Dear authors,
> > >
> > > Thank you for the clarifications and for incorporating Appendix F into the paper.  In light of these changes, I will revise my initial score.
> > >
> > > However, to facilitate a clearer connection between the main text and the added material, I recommend including a reference to Appendix F at the beginning of Section 2. This would enhance the coherence of your paper and assist readers in navigating the supplementary content.

---

> > > > ### Author Response · Authors · 2023-11-20
> > > >
> > > > Dear Reviewer 4dCg,
> > > >
> > > > Thank you for increasing you score. Your feedback is greatly appreciated and we will include a reference to Appendix F at the beginning of Section 2.